# Rapidly Adapting Policies to the Real World via Simulation-Guided Fine-Tuning

**Patrick Yin**[*,1], **Tyler Westenbroek**[*,1],
**Ching-an Cheng**[2], **Andrey Kobolov**[2] & **Abhishek Gupta**[1]

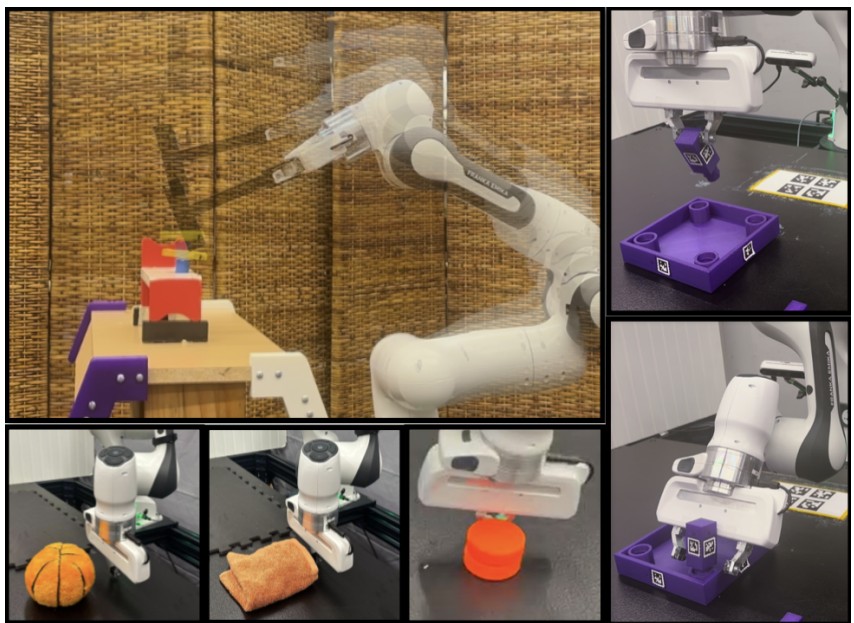

Figure 1: Five dynamic, contact-rich manipulation tasks – hammering (**top left**), insertion (**right**), and three pushing (**bottom left**) tasks – solved in the real world using *SGFT*.

## Abstract

Robot learning requires a considerable amount of high-quality data to realize the promise of generalization. However, large data sets are costly to collect in the real world. Physics simulators can cheaply generate vast data sets with broad coverage over states, actions, and environments. However, physics engines are fundamentally misspecified approximations to reality. This makes direct zero-shot transfer from simulation to reality challenging, especially in tasks where precise and force-sensitive manipulation is necessary. Thus, fine-tuning these policies with small real-world data sets is an appealing pathway for scaling robot learning. However, current reinforcement learning fine-tuning frameworks leverage general, unstructured exploration strategies which are too inefficient to make real-world adaptation practical. This paper introduces the *Simulation-Guided Fine-tuning* (SGFT) framework, which demonstrates how to extract structural priors from physics simulators to substantially accelerate real-world adaptation. Specifically, our approach uses a value function learned in simulation to guide real-world exploration. We demonstrate this approach across five real-world dexterous manipulation tasks where zero-shot sim-to-real transfer fails. We further demonstrate our framework substantially outperforms baseline fine-tuning methods, requiring up to an order of magnitude fewer real-world samples and succeeding at difficult tasks where prior approaches fail entirely. Last but not least, we provide theoretical justification for this new paradigm which underpins how SGFT can rapidly learn high-performance policies in the face of large sim-to-real dynamics gaps. Project webpage: weirdlabuw.github.io/sgft

[*]Equal contribution 1. University of Washington 2. Microsoft Research

# 1 INTRODUCTION

Robot learning offers a pathway to building robust, general-purpose robotic agents that can rapidly adapt their behavior to new environments and tasks. This shifts the burden from designing task-specific controllers by hand to the problem of collecting large behavioral datasets with sufficient coverage. This raises a fundamental question: *how do we cheaply obtain and leverage such datasets at scale?* Real-world data collection via teleoperation (Walke et al., 2023; team, 2024) can generate high-quality trajectories but scales linearly with human effort. Even with community-driven teleoperation (Mandlekar et al., 2018; Collaboration, 2024), current robotics datasets are orders of magnitude smaller than those powering vision and language applications.

Massively parallelized physics simulation (Todorov et al., 2012; Makoviychuk et al., 2021) can cheaply generate vast synthetic robotics data sets. Indeed, generating data sets with extensive *coverage* over environments, states, and actions can be largely automated using techniques such as automatic scene generation (Chen et al., 2024; Deitke et al., 2022), dynamics randomization (Peng et al., 2018; Andrychowicz et al., 2020), and search algorithms such as reinforcement learning.

Unfortunately, simulation-generated data is not a silver bullet, as it provides cheap but ultimately *off-domain data*. Namely, simulators are fundamentally *misspecified* approximations to reality. This is highlighted in tasks like hammering in a nail, where the modeling of high-impact, deformable contact remains an open problem (Acosta et al., 2022; Levy et al., 2024). In these regimes, *no choice of parameters for the physics simulator accurately capture the real-world dynamics*. This gap persists despite efforts towards improving existing physics simulators with system identification (Memmel et al., 2024; Huang et al., 2023; Levy et al., 2024). Thus, despite impressive performance for many tasks, methods that transfer policies from simulation to reality zero-shot (Kumar et al., 2021; Lee et al., 2020; Peng et al., 2018; Andrychowicz et al., 2020) run into failure modes when they encounter novel dynamics not covered by the simulator (Smith et al., 2022b).

The question becomes: *can inaccurate simulation models be useful in the face of fundamental misspecifications?* A natural technique is to fine-tune policies pre-trained in a simulator using real-world experience (Smith et al., 2022b; Zhang et al., 2023). This approach can overcome misspecification by training directly on data from the target domain. However, existing approaches typically employ the unstructured exploration strategies used by standard, general reinforcement learning algorithms Haarnoja et al. (2018). As a result, current RL fine-tuning frameworks remain too sample-inefficient for real-world deployment. We argue the following: despite getting the finer details wrong, physics simulators capture the rough structure of real-world dynamics well enough to *transfer targeted exploration and adaptation strategies from simulation to reality*.

This motivates the *Simulation-Guided Fine-Tuning* (SGFT) framework, which uses a value function $V_{sim}$ learned in the simulator to transfer behavioral priors from simulation to reality. Our key insight is that the ordering defined by $V_{sim}$ captures successful behaviors – such as reaching towards and object, picking it, and moving it to a desired position – which are invariant across simulation and reality, even if the low-level dynamics diverge substantially in the two domains. In more detail, *SGFT* departs from standard finetuning strategies by 1) using $V_{sim}$ to synthesize dense rewards to guide targeted real-world exploration and 2) shortening the learning horizon when adapting in the real-world. Prior approaches Smith et al. (2022b); Zhang et al. (2023) optimize the same infinite-horizon objective in both simulation and reality, and thus *only use the simulator to initialize real-world learning*. These approaches often suffer from *catastrophic forgetting* Wołczyk et al. (2024), where there is a substantial drop in policy performance early in the fine-tuning phase. Incorporating $V_{sim}$ into the reward enables the agent to retain structural information about the optimal behaviors learned in simulation throughout the fine-tuning processes, while shorter learning horizons are well-known to yield more trackable policy optimization problems Westenbroek et al. (2022); Cheng et al. (2019). Altogether, *SGFT* provides a stronger learning signal which enables base policy optimization algorithms to consistently and rapidly improve real-world performance.

Although *SGFT* is a very general framework for sim-to-real fine-tuning, we place a special emphasis on implementations which use sample efficient model-based reinforcement learning algorithms Janner et al. (2019); Hansen et al. (2024). Model-based approaches typically struggle with long-horizon tasks due to compounding errors in model predictions Janner et al. (2019), but our horizon-shortening strategy conveniently side-steps this challenge, fully unlocking the performance benefits promised by generative world models. We outline our contributions as follows:

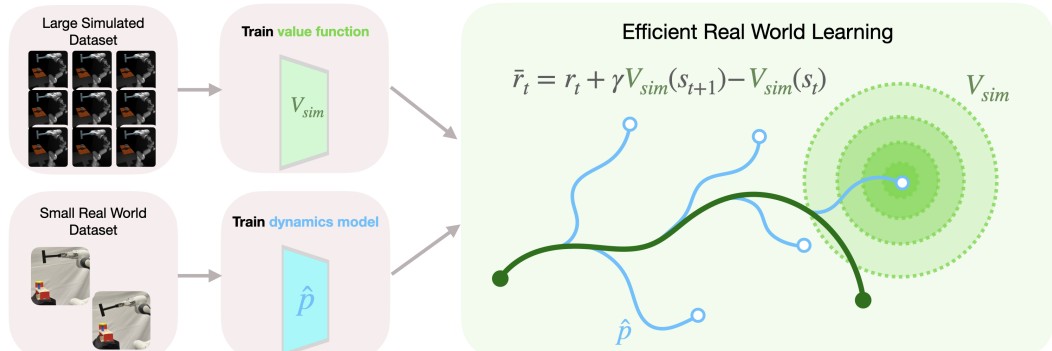

Figure 2: Depiction of a model-based instantiation of *SGFT*. While prior approaches optimize the same challenging infinite-horizon objective during simulation pretraining and real-world fine-tuning, *SGFT* modifies the fine-tuning objective by $a$) using the value function learned in simulation $V_{sim}$ to reshape rewards and guide efficient real-world exploration $b$) shortening the search horizon to make real-world learning more tractable. For the model-based instantiation depicted (and outlined in Section 4.2), this amounts to branching short, exploratory model-based rollouts from real-world trajectories to search for sequences of actions which will improve policy performance. Intuitively, *SGFT* uses the simulator to approximately bootstrap long-horizon rewards (via $V_{sim}$), while small amounts of real-world data are used to search for short sequences of optimal actions under the real-world dynamics.

1. We introduce the *SGFT* framework, which requires modifying only a few lines of code in existing sim-to-real RL finetuning frameworks Smith et al. (2022b); Zhang et al. (2019) and can be built on top of *any base policy optimization algorithm*.

2. We implement two model-based instantiations of *SGFT* and demonstrate through five contact-rich, sim-to-real robot manipulation experiments that *SGFT* provides substantial sample complexity gains over existing fine-tuning methods.

3. We demonstrate theoretically how $a$) *SGFT* can learn highly performant policies, despite the bias introduced by the *SGFT* objective and $b$) how this enables *SGFT* to overcome compounding errors which plague standard MBRL approaches.

Together, these insights underscore how *SGFT* effectively leverages plentiful off-domain data from a simulator alongside small real-world data sets to substantially accelerate real-world fine-tuning.

## 2 RELATED WORK

**Simulation-to-Reality Transfer.** Two main approaches have been proposed for overcoming the dynamics gap between simulation and reality: 1) adapting simulation parameters to real-world data (Chebotar et al., 2019; Ramos et al., 2019; Memmel et al., 2024) and 2) learning adaptive or robust policies to account for uncertain real-world dynamics (Qi et al., 2022; Kumar et al., 2021; Yu et al., 2017). However, these approaches still display failure modes in regimes where simulation and reality diverge substantially Smith et al. (2022a) – namely, where no set of simulation parameters closely match the real-world dynamics.

**Adapting Policies with Real-World Data.** Many RL approaches have focused on the general fine-tuning problem (Rajeswaran et al., 2018; Nair et al., 2020; Kostrikov et al., 2021; Hu et al., 2023; Nakamoto et al., 2024). These works initialize policies, Q-functions, and replay buffers from offline data and continue training them with standard RL methods. A number of works have specifically considered mixing simulated and real data during policy optimization – either through co-training (Torne et al., 2024), simply initializing the replay-buffer with simulation data (Smith et al., 2022a; Ball et al., 2023), or adaptively sampling the simulated dataset and up-weighting transitions that approximately match the true dynamics (Eysenbach et al., 2020; Liu et al., 2022; Xu et al., 2023; Niu et al., 2022). However, recent work Zhou et al. (2024) has demonstrated that there are minimal benefits to sampling off-domain samples when adapting online in new environments, as this can bias learned policies towards sub-optimal solutions. In contrast to these prior approaches – which primarily use simulated experience to initialize real-world learning – we focus on distilling effective exploration strategies from simulation, using $V_{sim}$ to guide real-world learning. We demonstrate theoretically that this approach to transfer leads to low bias in the policies learned by *SGFT*.

**Model-Based Reinforcement Learning.** A significant body of work on model-based RL learns a generative dynamics models to accelerate policy optimization (Sutton, 1991; Wang & Ba, 2019; Janner et al., 2019; Yu et al., 2020; Kidambi et al., 2020; Ebert et al., 2018; Zhang et al., 2019). In principle, a model enables the agent to make predictions about states and actions not contained in the training data, enabling rapid adaptation to new situations. However, the central challenge for model-based methods is that small inaccuracies in predictive models can quickly compound over time, leading to large *model-bias* and a drop in controller performance. An effective critic can be used to shorten search horizons (Hansen et al., 2024; Bhardwaj et al., 2020; Hafner et al., 2019; Jadbabaie et al., 2001; Grune & Rantzer, 2008) yielding easier decision-making problems, but learning such a critic from scratch can still require large amounts of on-task data. We demonstrate that, for many real-world continuous control problems, critics learned entirely in simulation can be robustly transferred to the real-world and substantially accelerate model-based learning.

**Reward Design in Reinforcement Learning.** A significant component of our methodology is learning dense shaped reward in simulation to guide real-world fine-tuning. Prior techniques have tried to infer rewards from expert demos (Ziebart et al., 2008; Ho & Ermon, 2016), success examples (Fu et al., 2018; Li et al., 2021), LLMs (Ma et al., 2023; Yu et al., 2023), and heuristics (Margolis et al., 2024; Berner et al., 2019). We rely on simulation to provide reward supervision Westenbroek et al. (2022) using the PBRS formalism (Ng et al., 1999). This effectively encodes information about the dynamics and optimal behaviors in the simulator, enabling us to shorten the learning horizon and improve sample efficiency Cheng et al. (2021); Westenbroek et al. (2022).

## 3 PRELIMINARIES

Let $\mathcal{S}$ and $\mathcal{A}$ be state and action spaces. Our goal is to control a real-world system defined by an unknown Markovian dynamics $s' \sim p_{real}(\cdot|s,a)$, where $s, s' \in \mathcal{S}$ are states and $a \in \mathcal{A}$ is an action. The usual formalism for solving tasks with RL is to define a Markov Decision Process of the form $\mathcal{M}_r = (\mathcal{S}, \mathcal{A}, p_{real}, \rho_{real}^0, r, \gamma)$ with initial real-world state distribution $\rho_{real}^0$, reward function $r$, and discount factor $\gamma \in [0,1)$. Given a policy $\pi$, we let $d_{real}^\pi(s)$ denote the distribution over trajectories $\tau = (s_0, a_0, s_1, a_1, \dots)$ generated by applying $\pi$ starting at initial state $s_0$. Defining the value function under $\pi$ as $V_{real}^\pi(s) = \mathbb{E}_{s_t \sim d_{real}^\pi(s)}[\sum_t \gamma^t r_t(s_t)]$, our objective is to find $\pi_{real}^* \leftarrow \sup_\pi \mathbb{E}_{s \sim \rho_{real}^0}[V_{real}^\pi(s)]$. We define the optimal value function as $V_{real}^*(s) := \sup_\pi V_{real}^\pi(s)$.

Unfortunately, obtaining a good approximation to $\pi_{real}^*$ using only real-world data is often impractical. Thus, many approaches leverage an approximate simulation environment $s' \sim p_{sim}(s,a)$ and solve an approximate MDP of the form $\mathcal{M}_{sim} := (\mathcal{S}, \mathcal{A}, p_{sim}, \rho_{sim}^0, r, \gamma)$ to train a policy $\pi_{sim}$ meant to approximate $\pi_{real}^*$. We let $V_{sim}$ denote $\pi_{sim}$'s value function with respect to $\mathcal{M}_{sim}$. Here, $\rho_{sim}^0$ is the distribution over initial conditions in the simulator.

## 4 SIMULATION-GUIDED FINE-TUNING

We build our framework around the following intuition: even when the finer details of $p_{real}$ and $p_{sim}$ differ substantially, we can often assume that $\pi_{sim}$ captures the rough motions needed to complete a task in the real world (such as swinging a hammer towards a nail). Specifically, we hypothesize that the *ordering* defined by $V_{sim}$ captures these behaviors in a form that can be robustly transferred from simulation to reality and used to guide efficient real-world adaptation.

### 4.1 SIMULATION-GUIDED FINE-TUNING

When fine-tuning $\pi_{sim}$ to the real world we propose $a)$ reshaping the original reward function according to $r(s) \to \bar{r}(s,s') = r(s) + \gamma V_{sim}(s') - V_{sim}(s)$ and $b)$ shortening the search horizon to a more tractable $H$-step objective $\sum_{t=0}^{H-1} \bar{r}(s_t, s_{t+1})$. The reshaped objective is an instance of the Potential-Based Reward Shaping (PRBS) formalism (Ng et al., 1999), where $V_{sim}$ is used as the *potential function*. This reshaping approach is typically applied to infinite-horizon objectives, where it can be shown that the optimal policy under $\bar{r}(s,s')$ is the same as the optimal policy under the original reward $r(s)$ (Ng et al., 1999). In contrast, the finite horizon search problem biases the objective towards behaviors which were successful in the simulator. Indeed, by telescoping out terms we can rewrite our policy optimization objective as:

$$V_H^{\pi_H}(s) = \mathbb{E}\left[\gamma^H V_{sim}(s_H) + \sum_{t=0}^{H-1} \gamma^t r(s_t) - V_{sim}(s_0) \Big| s_0 = s, a_t \sim \pi_{H,t}(\cdot|s_t)\right], \qquad (1)$$

$$V_H^*(s) := \sup_{\pi_H} V_H^{\pi_H}(s), \quad Q_H^*(s, \pi) := \mathbb{E}_{a \sim \pi(\cdot|s)}\big[\gamma V_H^*(s') + r(s)\big].$$

Here $\pi_H = \{\pi_{H,0}, \pi_{H,1}, \dots, \pi_{H,H-1}\}$ denotes $H$-step time-dependent policies, where $\pi_{H,t}$ is the policy applied at time $t$. We emphasize that these $H$-step returns are optimized over the *real-world dynamics*. We propose learning a policy which optimizes these $H$-step returns from each state:

$$\pi_H^*(\cdot|s) \leftarrow \sup_\pi Q_H^*(s, \pi).$$

**What behavior does this objective elicit?** For each $H$, the $H$-step Bellman equation dictates that the optimal policy $\pi_{real}^*$ for the original MDP $\mathcal{M}_{real}$ can be found by solving:

$$\pi_{real}^*(\cdot|s) \leftarrow \sup_\pi \mathbb{E}\left[\gamma^H V_{real}^*(s_H) + \sum_{t=0}^{H-1} \gamma^t r(s_t) - V_{real}^*(s_0)\,\bigg|\, s_0 = s, a_t \sim \pi(\cdot|s_t)\right].$$

Thus, (1) effectively uses $V_{sim}$ as a surrogate for $V_{real}^*$. Namely, (1) uses the simulator to bootstrap long-horizon returns, while real-world interaction is optimized only over short trajectory segments. In the extreme case where $H = 1$, $\pi_H^*$ will greedily attempt to increase $V_{sim}$ at each timestep; namely, the policy search problem will be reduced to a contextual bandit problem. As we take $H \rightarrow \infty$, $\pi_H^*$ will optimize purely real long-horizon returns and thus recover the behavior of $\pi_{real}^*$.

We are particularly interested in optimizing this objective with small values of $H$. This provides an ideal separation between what is learned using cheap, plentiful interactions with $p_{sim}$ and what is learned with more costly interactions with $p_{real}$. In simulation, we can easily generate enough data to explore many paths through the state space and discover which motions lead to higher returns (e.g. swinging a hammer towards a nail). This information is distilled into $V_{sim}$ during the learning process, which defines an ordering over which states are more desirable to reach $H$ steps in the future. By optimizing the reshaped objective Equation (1) for small values of $H$, we only need to learn short sequences of actions which move the real-world system to states where $V_{sim}$ is higher. Intuitively, (1) learns *where to go* with large amounts of simulated data and *how to get there* with small amounts of real-world data, efficiently adapting $\pi_{sim}$ to the real-world dynamcis.

**Connections to the MPC Literature:** Note that the $-V_{sim}(s_0)$ term in (1) does not depend on the choice of policy, and thus does not affect the choice of optimal policy. Thus, (1) is equivalent to the planning objective used by model predictive control (MPC) methods (Jadbabaie et al., 2001; Hansen et al., 2024; Sun et al., 2018; Bhardwaj et al., 2020) with $H$-step look-ahead and a terminal reward of $V_{sim}$ (assuming oracle access to the real-world dynamics). Of course, we cannot calculate the optimal MPC policy $\pi_H^*$ directly because we do not know $p_{real}$, and thus seek to approximate its behavior by learning from real-world interactions.

**The Simulation-Guided Fine-Tuning Training Loop.** We propose the general *Simulation-Guided Fine-Tuning (SGFT)* framework in the pseudo-code in Algorithm 1. *SGFT* fine-tunes $\pi_{sim}$ to succeed under the real-world dynamics by iteratively 1) unrolling the current policy to collect transitions from $p_{real}$ and 2) using the current dataset $\mathcal{D}$ of transitions to approximately optimize $\pi \leftarrow \max_\pi Q_H^*(s, \pi(s_j))$ at each state $s_j$ the agent has visited. By optimizing (1), *SGFT* optimizes policies towards the actions taken by $\pi_H^*$. Note that this framework can be built on top of any base policy

---

**Algorithm 1** Simulation-Guided Fine-tuning ( *SGFT*)

**Require:** Pretrained policy $\pi_{sim}$ and value function $V_{sim}$
1: $\pi \leftarrow \pi_{sim}, \mathcal{D} \leftarrow \emptyset$
2: **for** each iteration k **do**
3:     **for** time step $t = 1, \dots, T$ **do**
4:         $a_t \sim \pi(\cdot|s_t)$
5:         Observe the state $s_{t+1}$ and the reward $r_t$.
6:         $\bar{r}_t \leftarrow r_t + \gamma V_{sim}(s_{t+1}) - V_{sim}(s_t)$
7:         $\mathcal{D} \leftarrow \mathcal{D} \cup \{(s_t, a_t, \bar{r}_t, s_{t+1})\}$
8:     **end for**
9:     Approx. optimize $\pi \leftarrow \max_\pi Q_H^*(s, \pi(s_j))$
        using transitions in $\mathcal{D}$, at all observed states $s_j$.
10: **end for**

---

optimization method; however, over the next two sections we will argue that this framework is particularly beneficial when paired with model-based search strategies.

## 4.2 LEVERAGING SHORT MODEL ROLL-OUTS

Learning a generative model $\hat{p}$ for $p_{real}$ with the real-world data set $\mathcal{D}$ enables an agent to generate synthetic rollouts and reason about trajectories not contained in the data set. In principle, this should

substantially accelerate the learning of effective policies. The central challenge for model-based reinforcement learning is that small errors in $\hat{p}$ can quickly compound over multiple steps, degrading the quality of the predictions Janner et al. (2019). As a consequence, learning a model accurate enough to solve long-horizon problems can often take as much data as solving the task with modern model-free methods (Chen et al., 2021; Hiraoka et al., 2021). By bootstrapping $V_{sim}$ in simulation, where data is plentiful, the *SGFT* framework enables agents to act effectively over long horizons using only short, local $H$-step predictions about the real-world dynamics. This side-steps the core challenge for model-based methods, and enables extremely efficient real-world learning. We now outline how the SGFT can be built on top of two predominant classes of MBRL algorithms.

**Notation.** In what follows, we will use 'hat' notation to denote $H$-step returns of the form (1) under the transitions generated by the model $\hat{p}$ rather than the real dynamics $p_{real}$. Namely, $\hat{V}_H^{\pi_H}$ is the $H$-step value of policy $\pi_H$ under $\hat{p}$; $\hat{V}_H^*$ and $\hat{Q}_H^*$ are the optimal values. Similarly, $\hat{\pi}_H^*$ denotes the optimal $H$-step policy under the model dynamics. Note that this is simply the MPC policy generated by using $\hat{p}$ and optimizing (1) with $\hat{p}$ substituted in for $p_{real}$.

**Improved Sample Efficiency with Data Augmentation (Algorithm 2).** The generative model $\hat{p}$ can be used for *data augmentation* by generating a dataset of synthetic rollouts $\hat{\mathcal{D}}$ to supplement the real-world dataset $\mathcal{D}$ (Janner et al., 2019; Sutton, 1990; Gu et al., 2016). The combined dataset can then be fed to any policy optimization strategy, such as generic model-free algorithms. We consider state-of-the-art Dyna-style algorithms (Janner et al., 2019), which, in

---

**Algorithm 2** Dyna-*SGFT*

**Require:** Pretrained policy $\pi_{sim}$ and value function $V_{sim}$
1: $\pi \leftarrow \pi_{sim}$
2: **for** each iteration k **do**
3:     Generate rollout $\{(s_t, a_t, r_t, s_{t+1})\}_{t=0}^T$ under $\pi$.
4:     $\bar{r}_t \leftarrow r_t + \gamma V_{sim}(s_{t+1}) - V_{sim}(s_t)$
5:     $\mathcal{D} \leftarrow \mathcal{D} \cup (s_t, a_t, \bar{r}_t, s_{t+1})$
6:     Fit generative model $\hat{p}$ with $\mathcal{D}$.
7:     **for** G policy updates **do**
8:         Generate synthetic branched rollouts $\hat{\mathcal{D}}$ under $\pi$.
9:         Approx. optimize $\pi \leftarrow \max_\pi Q_H^*(s, \pi(s_j))$
            $\forall s_j \in \mathcal{D}$ using augmented dataset $\hat{\mathcal{D}} \cup \mathcal{D}$
10:     **end for**
11: **end for**

---

**Algorithm 3** MPC-*SGFT*

**Require:** Pretrained value $V_{sim}$ and initialized model $\hat{p}$.
1: **for** each iteration k **do**
2:     Generate rollout $\{(s_t, a_t, r_t, s_{t+1})\}_{t=0}^T$ using $\hat{p}$ and trajectory optimization to calculate $\hat{\pi}_H^*$
3:     $\bar{r}_t \leftarrow r_t + \gamma V_{sim}(s_{t+1}) - V_{sim}(s_t)$
4:     $\mathcal{D} \leftarrow \mathcal{D} \cup \{(s_t, a_t, \bar{r}_t, s_{t+1})\}$.
5:     Fit generative model $\hat{p}$ with $\mathcal{D}$.
6: **end for**

---

our context, branch $H$-step rollouts from states the agent has visited previously in the real-world. As Algorithm 2 shows, after each data-collection phase, this approach updates $\hat{p}$ and then repeatedly a) generates a dataset $\hat{\mathcal{D}}$ of synthetic $H$-step rollouts under the current policy $\pi$ starting from states in $\mathcal{D}$, and b) approximately solves $\pi \leftarrow \max_{\bar{\pi}} Q_H^*(s, \bar{\pi}(s))$ at the observed real-world states using the augmented dataset $\hat{\mathcal{D}} \cup \mathcal{D}$ and a base model-free method such as SAC (Haarnoja et al., 2018).

**Online Planning (Algorithm 3).** The most straightforward way to approximate the behavior of $\pi_H^*$ is simply to apply the MPC controller $\hat{\pi}_H^*$ generated using the current best guess for the dynamics $\hat{p}$. Algorithm 3 provides general pseudocode for this approach, which iteratively 1) rolls out $\hat{\pi}_H^*$ (which is calculated using online optimization and $\hat{p}$ (Williams et al., 2017)) then 2) updates the model on the current dataset of transitions $\mathcal{D}$. This high-level approach encompasses a wide array of methods proposed in the literature, e.g., Ebert et al. (2018) and Zhang et al. (2019). For the experiments in Section 6, we implemented this approach using the TDMPC-2 (Hansen et al., 2024) algorithm, which additionally learns a policy prior to accelerate the trajectory optimization.

## 5 THEORETICAL ANALYSIS

The preceding discussions have covered how the dense rewards and horizon shortening strategy employed by *SGFT* can lead to efficient real-world adaptation. However this leaves the outstanding question: *how does the bias introduced by the reward shaping and horizon shortening affect policy performance?* Longer prediction horizons decrease the bias of the objective by relying more heavily on real returns, but lead to less sample efficient adaptation Cheng et al. (2021); Westenbroek et al.

(2022). Given these competing challenges, can we expect *SGFT* to simultaneously learn high-performing policies and adapt rapidly in the real-world?

We introduce a novel geometric analysis which demonstrates that *SGFT* can achieve both goals, even when there is a large gap between $p_{sim}$ and $p_{real}$. Specifically, we introduce mild technical conditions on the structure of $p_{sim}$, $V_{sim}$ and $p_{real}$ which ensure that $\pi_H^*$ is nearly optimal for $\mathcal{M}_{real}$, even when short prediction horizons $H$ are used. Our analysis builds on the following insight: even when there is a large gap in the magnitude $|p_{sim} - p_{real}|_\infty$ between simulation and reality, it is reasonable to assume that $V_{sim}$ still defines a reasonable *ordering* over desirable states under the real-world dynamics $p_{real}$. Namely, we argue that $V_{sim}$ can capture the structure of motions that complete the desired task (such as reaching towards and object, picking it up, and moving it to a desired location), even if the low-level sequences of actions needed to realize these motions differs substantially between simulation and reality (due e.g. to difficult-to-model contact dynamics). We use the following formal definition to capture this intuition, which is from Cheng et al. (2019) and shares strong connections to Lyapunov Westenbroek et al. (2022); Grune & Rantzer (2008) and Dissipation Brogliato et al. theories from dynamical systems and control:

**Definition 1.** *We say that $V_{sim}$ is* improvable *with respect to $\mathcal{M}_{real}$ if for each $s \in \mathcal{S}$ we have:*

$$\max_a \mathbb{E}_{s' \sim p_{real}(\cdot|s,a)}[\gamma V_{sim}(s')] - V_{sim}(s) \geq -r(s). \tag{2}$$

To unpack why this definition is useful, note that $V_{sim}$ is by definition improvable with respect to $\mathcal{M}_{sim}$. Indeed, by the temporal difference equation we have $\mathbb{E}_{s' \sim p_{sim}(\cdot|s,a),\, a \sim \pi_{sim}(\cdot|s)}[\gamma V_{sim}(s')] - V_{sim}(s) = -r(s)$. Namely, $V_{sim}$ is constructed so that polices can greedily increase $V_{sim}$ under $p_{sim}$ at each step and be guaranteed to reach desirable states in the long run, namely, states which correspond to successfully completing the task. Definition 1 ensures that this condition then holds *under the real-world dynamics*; this requirement ensures that the ordering defined by $V_{sim}$ encodes feasible motions that solve the task in the real-world. This enables policy optimization algorithms to greedily follow $V_{sim}$ at each step while ensuring that the resulting behavior successfully completes the task. We use the following pedagogical example to investigate why this is a reasonable property to assume for continuous control problems:

**Pedagogical Example.** Consider the following case where the real and simulated dynamics are both deterministic, namely, $s' = p_{real}(s,a)$ and $s' = p_{sim}(s,a)$. Specifically, consider the case where $s = (s_1, s_2) \in \mathcal{S} \subset \mathbb{R}^2$, $a \in \mathcal{A} = \mathbb{R}$, and the dynamics are given by:

$$p_{sim}(s,a) = \begin{bmatrix} s_1' \\ s_2' \end{bmatrix} = \begin{bmatrix} s_1 \\ s_2 \end{bmatrix} + \Delta t \begin{bmatrix} s_2 \\ \frac{g}{l}\sin(s_1) + a \end{bmatrix}$$

$$p_{real}(s,a) = \begin{bmatrix} s_1' \\ s_2' \end{bmatrix} = \begin{bmatrix} s_1 \\ s_2 \end{bmatrix} + \Delta t \begin{bmatrix} s_2 \\ \frac{g}{l}\sin(s_1) + a + e(s_1, s_2) \end{bmatrix}.$$

These are the equations of motion for a simple pendulum (Wang et al., 2022) under an Euler discretization with time step $\Delta t$, where $s_1$ is the angle of the arm, $s_2$ is the angular velocity, $a$ is the torque applied by the motor, $g$ is the gravitational constant, and $l$ is the length of the arm. The real-world dynamics contains unmodeled terms $e(s_1, s_2)$, which might correspond to complex frictional or damping effects. Consider the policy for the real world given by $\pi_{real}(s) = \pi_{sim}(s) - e(s_1, s_2)$ and observe that $p_{sim}(s, \pi_{sim}(s)) = p_{real}(s, \pi_{real}(s))$. Even though the difference in transition dynamics $e(s_1, s_2)$ might be quite large, the set of feasible next states is the same for the two environments. That is, the space of feasible motions in the two MDPs are identical, even though it takes *substantially different policies to realize these motions*.

**Geometric Insight.** More broadly, if for each $s$ there exists some $a$ such that $p_{real}(\cdot|s,a) = p_{sim}(\cdot|s, \pi_{sim}(s))$, then $V_{sim}$ is improvable with respect to $\mathcal{M}_{real}$. This follows from the fact that $V_{sim}$ is improvable under $p_{sim}$ by definition (see above discussion) and the fact that there exists actions which exactly match the state transitions in simulation and reality. More generally, it is reasonable to expect that $p_{sim}$ approximately captures the geometry of what motions are feasible under $p_{real}$, even if the actions required to realize the motions differ substantially in the two MDPs. Thus it is reasonable to assume $V_{sim}$ is improvable. This intuition is highlighted by our real-world examples in cases where we use *SGFT* with a prediction horizon of $H = 1$. In these cases the learned policy is able to greedily follow $V_{sim}$ at each state and reach the goal, even in the face of large dynamics gaps. Relatedly, work on Lyapunov theory Westenbroek et al. (2022) has demonstrated theoretically that value functions are naturally robust to dynamics shifts under mild conditions.

**Main Theoretical Result.** Before presenting our main theoretical result, we provide a useful point of comparison from the literature Bhardwaj et al. (2020). When translated to our setting, [1] (Bhardwaj et al., 2020, Theorem 3.1) assumes that $\forall (s, a)$ we have $a)$ the value gap between simulation and reality is bounded by $\|V_{sim}(s) - V^*_{real}(s)\| < \epsilon$ and $b)$ that a generative model $\hat{p}$ is used for planning wherein $\|\hat{p}(\cdot|s, a) - p_{real}(\cdot|s, a)\|_1 < \alpha$. Recalling that $\hat{\pi}^*_H$ is the controller synthesized using the model $\hat{p}$, (Bhardwaj et al., 2020, Theorem 3.1) bounds the suboptimality for the model-based controller as $V^*_{real}(s) - V^{\hat{\pi}^*_H}_{real}(s) \leq O\left(\frac{\gamma}{1-\gamma}\alpha H + \frac{\gamma^H}{1-\gamma^H}\epsilon\right)$. To understand the bound, first set $\alpha = 0$ so that there is no modeling error (as in the case of model-free instantiations of *SGFT*). In this regime we are incentivized to increase $H$, as this will decrease the $\frac{\gamma^H}{1-\gamma^H}\epsilon$ term and improve performance. However this term scales very poorly for long-horizon problems where $\gamma \approx 1$, and we may need large values of $H$ to obtain near optimal policies. When $\alpha > 0$, the situation becomes even more challenging. Increasing $H$ will increase the $\frac{\gamma}{1-\gamma}\alpha H$ term, reflecting how longer prediction horizons propagate model-based errors and increase suboptimality. Thus, without additional structural assumptions, we cannot conclude that *SGFT* can learn high-performance controllers when $H$ is small. The following result demonstrates this is possible when $V_{sim}$ is improvable:

**Theorem 1.** *Assumes that $\forall (s, a)$ we have $a)$ the value gap between simulation and reality is bounded by $\|V_{sim}(s) - V^*_{real}(s)\| < \epsilon$ and $b)$ if a generative model $\hat{p}$ is used by SGFT then $\|\hat{p}(\cdot|s, a) - p_{real}(\cdot|s, a)\|_1 < \alpha$. Further suppose that $V_{sim}$ is* improvable *with respect to $\mathcal{M}_{real}$. Then for $H$ sufficiently small and each $s \in \mathcal{S}$ we have:*

$$V^*_{real}(s) - V^{\hat{\pi}^*_H}_{real}(s) \leq O\left(\frac{\gamma}{1-\gamma}\alpha H + \gamma^H \epsilon\right), \tag{3}$$

*where $\hat{\pi}^*_H$ is the policy learned by SGFT.*

See Appendix A for proof. This result demonstrates that when $V_{sim}$ is improvable with respect to $\mathcal{M}_{real}$ *SGFT* can obtain nearly optimal policies, even when $H$ is small. When compared to (Bhardwaj et al., 2020, Theorem 3.1), which assumes uniform worst-case bounds on $\|V_{sim} - V_{real}\|$, when $V_{sim}$ is improvable we gain a substantial factor of $\frac{1}{1-\gamma^H}$ when bounding the effects of errors in $V_{sim}$. In short, this result demonstrates that even mild structural consistency between $p_{sim}$ and $p_{real}$ is enough to ensure that *SGFT* provides effective guidance towards performant policies when $H$ is small. In the case of MBRL ($\alpha > 0$), this is especially important as keeping $H$ small combats compounding errors in the dynamics model. Our proof technique adapts ideas from the theoretical control literature Grune & Rantzer (2008); Westenbroek et al. (2022) to the more general setting we consider, and can be seen as a model-based analogy to the results from Cheng et al. (2021).

## 6 EXPERIMENTS

We answer the following: **(1)** Can *SGFT* facilitate rapid online fine-tuning for dynamic, contact-rich manipulation tasks? **(2)** Does *SGFT* improve the sample efficiency of fine-tuning compared to baselines? **(3)** Can *SGFT* learn successful policies where prior methods fail entirely?

### 6.1 METHODS EVALUATED

*SGFT* **Instantiations.** We implement concrete instantiations of the general Dyna-*SGFT* and MPC-*SGFT* frameworks sketched in Algorithms 2 and 3. *SGFT*-**SAC** fits a model to real world transitions to perform data augmentation and uses SAC as a base model-free policy optimization algorithm. We use $H = 1$ all our experiments. Crucially, we set the 'done' flag to true at the end of each rollout – this ensures SAC does not bootstrap its own critic from the real-world data and only uses $V_{sim}$ to bootstrap long-horizon returns. *SGFT*-**TDMPC-2** uses TDMPC-2 Hansen et al. (2024) as a backbone. The base method learns a critic, a policy, and an approximate dynamics model through interactions with the environment. When acting in the world, the MPC controller solves online planning problems using the approximate model, the critic as a terminal reward, and uses the policy prior to seed an MPPI planner Williams et al. (2017). To integrate this method with SGFT, when transferring to the real world we simply freeze the critic learned in simulation and use the reshaped objective in Equation (1) as the online planning objective. For our experiments, we use $H = 4$ and the default hyperparameters reported in Hansen et al. (2024).

---

[1] We note that Bhardwaj et al. (2020) focuses on general MPC problems where some approximation $\hat{V} \approx V^*_{real}$ is used as the terminal cost for the planning objective. However, despite the difference in settings, the structure of the underlying $H$-step objective we consider is identical to much of the MPC literature.

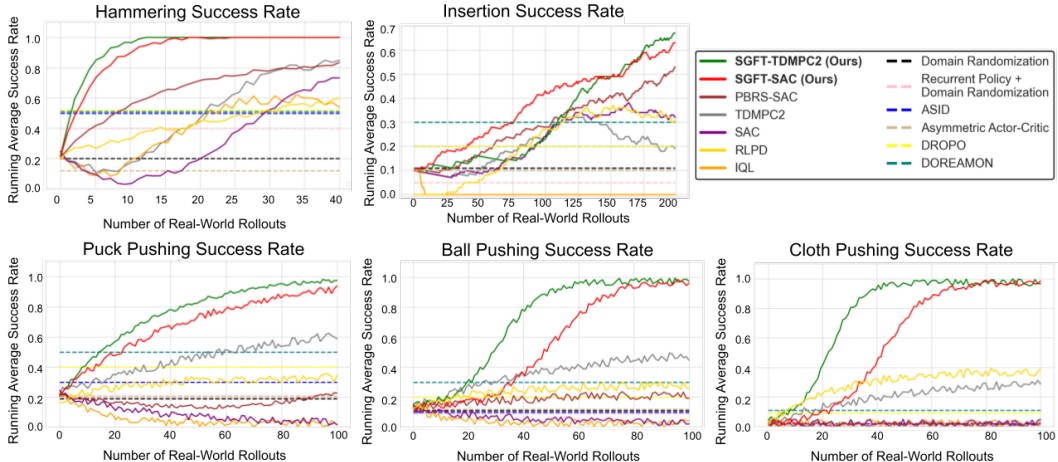

Figure 4: **Real-world success rates during the course of online fine-tuning.** We plot task success rates over number of fine-tuning rollouts for the tasks described in Sec. 6. We see that *SGFT* yields significant improvements in success and efficiency.

**Baseline Fine-tuning Methods.** We also fine-tune polices using the original infinite-horizon objective for $\mathcal{M}_{real}$, including **SAC** Haarnoja et al. (2018), **TDMPC-2** Hansen et al. (2024), **IQL** Kostrikov et al. (2021). **PBRS** fine-tunes the policy using the reshaped reward, an infinite horizon, and SAC (Haarnoja et al., 2018)– it does not use horizon shortening. **RLPD** fine-tunes a fresh policy to solve the original MDP $\mathcal{M}_{real}$ using RLPD (Ball et al., 2023). *These algorithm cover state-of-the-art model-based, model-free, and offline-to-online adaptation algorithms.*

**Baseline Sim-to-Real Methods.** Our **Domain Randomization** baseline refers to policies trained with extensive domain randomization in simulation and transferred directly to the real world. These policies rely only on the previous observation. **Recurrent Policy + Domain Randomization** uses policies conditioned on histories of observations, similar to methods such as Kumar et al. (2021). **Asymmetric Actor-Critic** (Pinto et al., 2018) uses policies trained in simulation with a critic conditioned on privileged information. **DOREAMON** (Tiboni et al., 2023b) is a recently proposed transfer method which automatically generates curricula to enable robust transfer. **ASID** (Memmel et al., 2024) and **DROPO** (Tiboni et al., 2023a) identify simulation parameters with small real-world data sets. *These methods serve as a state-of-the-art baselines for sim-to-real transfer methods.*

## 6.2 SIM-TO-REAL EVALUATIONS

We test each method on five real-world manipulation tasks illustrated in Figure 3 and two additional real-world deformable object pushing tasks illustrated in Figure 1, demonstrating that both the *SGFT*-SAC and *SGFT*-TDMPC-2 instantiations of *SGFT* excel at learning policies with minimal real-world data.

**Hammering** is a highly dynamic task involving force and contact dynamics that are impractical to precisely model in simulation. In our setting, the robot is tasked with hammering a nail in a board. The nail has high, variable dry friction along its shaft. In order to hammer the nail into the board, the robot must hit the nail with high force repeatedly. The dynamics are inherently

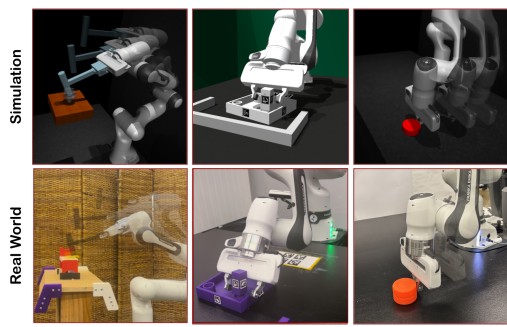

Figure 3: **Sim-to-Real Setup** Simulation setup for pre-training (**top**) and execution of real-world fine-tuning (**bottom**) of real-world hammering (**left**), insertion (**middle**), and pushing (**right**).

misspecified between simulation and reality here due to the infeasability of accurately modeling the properties of the nail and its contact interaction with the hammer and board.

**Insertion** (Heo et al., 2023) involves the robot grasping a table leg and accurately inserting it into a table hole. The contact dynamics between the leg and the table differ between simulation and real-world conditions. In the simulation, the robot successfully completes the task by wiggling the

leg into the hole, but in the real world this precise motion becomes challenging due to inherent noise in the real-world observations as well as contact discrepancies between the leg and the table hole.

**Puck Pushing** requires pushing a puck of unknown mass and friction forward to the edge of the table without it falling off the edge. Here, the underlying feedback controller of the real world robot inherently behaves differently from simulation. Additionally, retrieving and processing sensor information from cameras incurs variable amounts of latency. As a result, the controller executes each commanded action for variable amounts of time. These factors all contribute to the sim-to-real dynamics shift, requiring real-world fine-tuning to reconcile.

**Deformable Object Pushing** is similar to puck pushing but involves pushing deformable objects which are challenging to model in simulation. As a result, we choose to model the object as a puck in simulation, making the simulation dynamics fundamentally misspecified compared to the real world. This example exemplifies how value functions trained under highly different dynamics in simulation can still provide useful behavioral priors for real-world exploration. We include two deformable pushing tasks – a towel and a squishy toy ball.

Each task is evaluated on a physical setup using a Franka FR3 operating with either Cartesian position control or joint position control at 5Hz. We compute object positions by color-thresholding pointclouds or by Aruco marker tracking, although this approach could easily be upgraded. Further details on reward functions, robot setups and environments can be found in Appendix B.

**Analysis.** The results of real-world fine-tuning on these five tasks are presented in Figure 4. For all five tasks, zero-shot performance is quite poor due to the dynamics sim-to-real gap. The poor performance of direct sim-to-real transfer methods such as Domain Randomization, Recurrent Policy + Domain Randomization (Kumar et al., 2021), Asymmetric Actor-Critic (Pinto et al., 2018), ASID (Memmel et al., 2024), DROPO (Tiboni et al., 2023a), and DOREAMON (Tiboni et al., 2023b) highlight that these gaps are due to more than parameter misidentification or poor policy training in simulation, but rather stem from fundamental simulator misspecification.

The second class of comparison methods includes offline pretraining with online fine-tuning techniques like IQL (Kostrikov et al., 2021), SAC (Haarnoja et al., 2018), and RLPD (Ball et al., 2023). Whether model-free or model-based, the SGFT finetuning methods (ours) substantially outperform these techniques in terms of efficiency and asymptotic performance. Moreover, they prevent *catastrophic forgetting*, wherein finetuning leads to periods of sharp degradation in the policies effectiveness. This suggests that simulation can offer more guidance during real-world policy search than just network weight initialization and/or replay buffer data initialization for subsequent fine-tuning. Our full system consistently leads to significant improvement from fine-tuning, achieving 100% success for hammering and pushing within an hour of fine-tuning and 70% success for inserting within two hours of fine-tuning. The fact that SGFT outperforms both TD-MPC2 (Hansen et al., 2024) and PBRS-SAC, suggests that efficient fine-tuning requires a *combination* of both short model rollouts and value-driven reward shaping. Last but not least, note that SGFT offers improvements on top of both SAC and TDMPC2, showing the generality of the proposed paradigm.

In the Appendix we also include $a$) a set of sim-to-sim transfer benchmarks and hyper-parameter ablations (Appendix E) $b$) visualizations of transferred value functions (Appendix D).

## 7 LIMITATIONS AND FUTURE WORK

We present *SGFT*, a general framework for efficient sim-to-real fine-tuning with existing RL algorithms. The key idea is to leverage value functions learned in simulation to provide guidance for real-world exploration. In the future it will be essential to scale methods to work directly from perceptual inputs. Calculating dense rewards from raw visual inputs is challenging, and represents an important limitation of the current instantiation of *SGFT*. Future work will investigate which visual modalities lead to robust sim-to-real transfer. Moreover, despite the sample complexity gains *SGFT* affords existing RL algorithms, we believe new base adaptation algorithms designed specifically for low-data regimes will be needed to make truly autonomous real world adaptation practical. For all methods tested, we found that tuning for real-world performance was time-consuming. Thus, future work will investigate using large simulated data sets to distill novel policy optimization algorithms that can be transferred to the real-world with less effort.

## 8 ACKNOWLEDGMENTS

We would like to thank Marius Memmel, Vidyaaranya Macha, Yunchu Zhang, Octi Zhang, and Arhan Jain for their invaluable engineering help. We would like to thank Lars Lien Ankile and Marcel Torne for their fruitful engineering discussions and constructive feedbacks. Patrick Yin is supported by sponsored research with Hitachi and the Amazon Science Hub.

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

## A    PROOFS

**Notation Recap.**    We remind the reviewer of notation we have built up throughout the paper. We use the 'hat' notation to denote a generative dynamics model $\hat{p}$, as well that the optimal values $\hat{V}_H^*, \hat{Q}_H^*$ obtained by optimizing the $H$-step objective under these dynamics. $\hat{\pi}_H^*$ is then the policy obtained by optimizing the $H$-step objective under the model dynamics: $\hat{\pi}_H(\cdot|s) \leftarrow \max_\pi \hat{Q}_H^*(a, \pi)$.

We first present several Lemma's used in the proof of Theorem 1. The first result bounds the difference between the true $H$-step returns for a policy $\pi_H$ and the $H$-step returns predicted under the dynamics model $\hat{p}$.

**Lemma 1.** *(Bhardwaj et al., 2020, Lemma A.1.) Suppose that $\|\hat{p}(s, a) - p_{real}(s, a)\|_1 \leq \alpha$. Further suppose $\Delta r = \max_s r(s) - \min_s r(s)$ and $\Delta V = \max_s V_s(s) - \min_s V_s(s)$ are finite. Then, for each policy $\pi$ we may bound the $H$-step returns under the model and true dynamics by:*

$$\|\hat{V}_H^{\pi_H}(s) - V_H^{\pi_H}\|_\infty \leq \gamma \left( \frac{1 - \gamma^{H-1}}{1 - \gamma} \frac{\Delta r}{2} + \gamma^H \frac{\Delta V}{2} \right) \cdot \alpha H. \tag{4}$$

*Proof.* This result follows imediatly from the proof of (Bhardwaj et al., 2020, Lemma A.1.), with changes to notation and noting that we assume access to the true reward. In particular, the full result of (Bhardwaj et al., 2020, Lemma A.1.) includes an extra $\frac{1-\gamma^H}{1-\gamma}\alpha$ term which comes from the usage of a model $\hat{r}$ which estimates the true reward. We do not consider such effects, and thus suppress this dependence. □

The next result uses this 1 to bound the difference between the optimal $H$-step returns and the $H$-step returns generated by the policy $\hat{\pi}_H^*$ which is optimal under the dynamics model $\hat{p}$:

**Lemma 2.** *Suppose that $\|\hat{p}(s, a) - p_{real}(s, a)\|_1 \leq \alpha$. Further suppose $\Delta r = \max_s r(s) - \min_s r(s)$ and $\Delta V = \max_s V_s(s) - \min_s V_s(s)$ are finite. Then for each state $s \in \mathcal{S}$ we have:*

$$V_H^*(s) - V_H^{\hat{\pi}_H^*}(s) \leq \left( \frac{1 - \gamma^{H-1}}{1 - \gamma} \Delta r + \gamma^H \Delta V \right) \tag{5}$$

*where $\hat{\pi}_H^* \leftarrow \max_{\pi_H} \hat{V}^{\tilde{\pi}_H}(s)$.*

*Proof.* Let $\pi_H^* \leftarrow \max_{\pi_H} V_H^{\pi_H}(s)$ be the optimal $H$-step policy under the true dynamics. By Lemma 1 we have both that

$$V_H^*(s) \leq \hat{V}^{\pi_H^*}(s) + \gamma \left( \frac{1 - \gamma^{H-1}}{1 - \gamma} \frac{\Delta r}{2} + \gamma^H \frac{\Delta V}{2} \right) \cdot \alpha H. \tag{6}$$

$$\hat{V}_H^{\hat{\pi}_H^*}(s) \leq V_H^{\hat{\pi}_H^*}(s) + \gamma \left( \frac{1 - \gamma^{H-1}}{1 - \gamma} \frac{\Delta r}{2} + \gamma^H \frac{\Delta V}{2} \right) \cdot \alpha H. \tag{7}$$

Combining these two bounds with the fact that $\hat{V}^{\pi_H^*}(s) \leq \hat{V}_H^{\hat{\pi}_H^*}(s)$ yields the desired result. □

The following result establishes an important monotonicity property on the optimal $H$-step value functions which is important for the main result.:

**Lemma 3.** *Suppose that $\sup_a \mathbb{E}_{s \sim p_{real}(s,a)}[\gamma V_{sim}(s')] - V_{sim}(s) > -r(s)$. Then we have $V_H^*(s) \geq V_{H-1}^*(s)$ for each $s \in \mathcal{S}$. Then for each $s \in \mathcal{S}$ we have:*

$$V_H^*(s) \geq V_{H-1}^*(s) \tag{8}$$

*Proof.* Fix an initial condition $s_0 \in \mathcal{S}$. Let $\pi$ be arbitrary, and fix the shorthand $\pi^* = \{\pi_0^*, \dots, \pi_{H-1}^*\}$ for the time-varying policy $\pi^* \leftarrow \max_{\tilde{\pi}} V_{H-2}^{\hat{\pi}}(s_0)$. Then, concatenate these policies to define: $\bar{\pi} = \{\pi_1^*, \dots, \pi_{H-2}^*, \pi\}$, which is simply the result of applying the optimal policy for the $(H-1)$-step look ahead objective Equation (1) starting from $s_0$, followed by applying $\pi$ for a single step. Letting the following distributions over trajectories by generated by $\pi^*$, by the definition

of $V_H^*$:

$$V_H^*(s_0)$$

$$\geq \mathbb{E}\left[\gamma^H V_{sim}(s_H) + \sum_{t=1}^{H-1} \gamma^t r(s_t) - V_{sim}(s_0)\right]$$

$$= \mathbb{E}\left[\gamma^H V_{sim}(s_H) - \gamma^{H-1} V_{sim}(s_{H-1}) + \gamma^H r(s_{H-1})\right]$$

$$+ \mathbb{E}\left[\gamma^{H-1} V_{sim}(s_{H-1}) + \sum_{t=1}^{H-2} \gamma^t r(s_t) - V_{sim}(s_0)\right]$$

$$= \mathbb{E}\left[\gamma^H V_{sim}(s_H) - \gamma^{H-1} V_{sim}(s_{H-1}) + \gamma^H r(s_{H-1}) + V_{H-1}^*(s_0)\right]$$

The first inequality follows from the fact that the return of $\bar{\pi}$ cannot be greater than that of $\pi^*$, the first equality follows from rearanging terms to isolate $V_{H-1}^*$, and the second equality follows from the definition of $V_{H-1}^*$. Now, since our choice of $\pi$ used to define $\bar{\pi}$ was arbitrary, we choose $\pi$ to be deterministic and such that $\mathbb{E}_{s' \sim p_{real}(s,a)}[\gamma V_{sim}(s')] - V_{sim}(s) > -r(s)$ at each state $s \in \mathcal{S}$, as guaranteed by the assumption made for the result. This choice of policy grantees that:

$$\mathbb{E}\left[\gamma^H V_{sim}(s_H) - \gamma^{H-1} V_{sim}(s_{H-1}) + \gamma^H r(s_{H-1})\right] \geq 0. \tag{9}$$

The desired result follows immediately by combining the two preceding bounds, and noting that our choice of initial condition was arbitrary, meaning the preceding analysis holds for all initial conditions. $\square$

Our final lemma bounds the sub-optimality of *SGFT* policies $\pi$ in terms of $a$) errors in the sim value function and $b$) additional suboptimalities cause by $\pi$ being sub-optimal for the $H$-step objective:

**Lemma 4.** *Suppose that $V_{sim}$ is improvable and further suppose that $\max_{s \in \mathcal{S}} |V_{sim}(s) - V_{real}^*(s)| < \epsilon$. Then any policy $\pi$ which satisfies $A_H^*(s, \pi) = Q_H^*(s, \pi) - V_H^*(s) \geq -\delta$ will satisfy:*

$$V_{real}^*(s) - V_{real}^\pi(s) \leq \gamma^H \epsilon + \frac{\delta}{1 - \gamma}. \tag{10}$$

*Proof.* Our goal is first to bound how $Q_H^*(s, \pi)$ changes on expectation when applying the given policy for a single step. We have that:

$$Q_H^*(s, \pi) + \delta \geq V_H^*(s) \tag{11}$$

$$V_H^*(s) \geq V_{H-1}^*(s) \tag{12}$$

$$Q_H^*(s, \pi) = \mathbb{E}[\gamma V_{H-1}^*(s') + \bar{r}(s, s')] \tag{13}$$

where the first inequality follows from the ssumption of the theorem, the second inequality follows from Lemma 3 and, the third inequality is simply the definition of $Q_H^*$. Letting $s' \sim p_{real}(s, a)$ with $a \sim \pi(\cdot|s)$, we can take expectations can combine the previous relations to obtain:

$$\mathbb{E}\left[\gamma Q_H^*(s', \pi) + \bar{r}(s, s')\right] + \gamma \delta \geq \mathbb{E}\left[\gamma V_H^*(s') + \bar{r}(s, s')\right] + \geq \mathbb{E}\left[\gamma V_{H-1}^*(s') + \bar{r}(s, s')\right] = Q_H^*(s, \pi). \tag{14}$$

That is:

$$\gamma \mathbb{E}\left[Q_H^*(s', \pi)\right] + \bar{r}(s) + \gamma \delta \geq Q_H^*(s, \pi). \tag{15}$$

Alternatively:

$$\bar{r}(s) \geq Q_H^*(s, \pi) - \gamma \mathbb{E}[Q_H^*(s', \pi)] - \gamma \delta. \tag{16}$$

Next, we use this bound to provide a lower bound for $V_{real}^\pi(s)$. Because the previous analysis holds at all states when we apply $\pi$, the following holds over the distribution of trajectories generated by

applying $\pi$ starting from the initial condition $s_0$ :

$$\mathbb{E}_{\rho^{\pi}_{real}(s)}\left[\sum_{t=0}^{\infty}\gamma^t\bar{r}(s_t)\right] = V^{\pi}_{real}(s) - V_s(s_0)$$

$$\geq \mathbb{E}_{\rho^{\pi}_{real}(s)}\left[\sum_{t=0}^{\infty}\gamma^t\left(Q^*_H(s_t,\pi) - \gamma Q^*_H(s_{t+1},\pi)\right)\right] - \gamma\delta\sum_{t=0}^{\infty}\gamma^t$$

$$= Q^*_H(s_0,\pi) - \frac{\gamma\delta}{1-\gamma},$$

where we have repeatedly telescoped out sums to cancel out terms in the first equality, used (16) in the second equality, and canceled out terms to generate the final equality.

Thus, we have the lower-bound:

$$V^{\pi}_{real}(s) \geq Q^*_H(s,\pi) + V_{sim}(s_0) - \frac{\gamma\delta}{1-\gamma} \tag{17}$$

$$\geq V^*_H(s) + V_{sim}(s_0) - \frac{\gamma\delta}{1-\gamma}$$

Next, we may bound:

$$V^*_H(s_0) + V_{sim}(s_0) \geq \mathbb{E}_{\rho^{\pi^*_{real}}(s_0)}\left[\gamma^H V_{sim}(s_H) + \sum_{t=0}^{H-1}\gamma^t r(s_t)\right] \tag{18}$$

$$= \mathbb{E}_{\rho^{\pi^*_{real}}(s_0)}\left[\gamma^H V_{sim}(s_H) - \gamma^H V^*_{real}(s_H) + \gamma^H V^*_{real}(s_H) + \sum_{t=0}^{H-1}\gamma^t r(s_t)\right]$$

$$= \mathbb{E}_{\rho^{\pi^*_{real}}(s_0)}\left[\gamma^H V_{sim}(s_H) - \gamma^H V^*_{real}(s_H)\right] + V^*_{real}(s_0).$$

Invoking the assumption that $\max_s |V_{sim}(s) - V^*_{real}(s)| < \epsilon$, we can combined this with the preceding bound to yield:

$$V^*_H(s_0) + V_{sim}(s_0) \geq V^*_{real}(s) - \gamma^H\epsilon. \tag{19}$$

Finally, once more invoking the fact that $Q^*_H(s,\pi) + \delta \geq V^*_H(s)$ for each $s \in S$ and combining this with Equation (17) and Equation (18), we obtain that:

$$V^{\pi}_{real}(s) \geq Q^*_H(s,\pi) + V_{sim}(s_0) - \frac{\gamma\delta}{1-\gamma}$$

$$\geq V^*_H(s_0) + V_{sim}(s_0) - \frac{\gamma\delta}{1-\gamma} - \delta$$

$$\geq V^*_{real}(s) - \frac{\gamma\delta}{1-\gamma} - \delta - \gamma^H\epsilon$$

$$= V^*_{real}(s) - \frac{\delta}{1-\gamma} - \gamma^H\epsilon$$

from which the state result follows immediately. $\square$

**Proof of Theorem 1:**

*Proof.* The result follows directly from a combination of Lemma 4 and Lemma 2 by suppressing problem-dependent constants and lower order terms in the discount factor $\gamma$. $\square$

## B ENVIRONMENT DETAILS

**Sim2Real Environment.** We use a 7-DoF Franka FR3 robot with a 1-DoF parallel-jaw gripper. Two calibrated Intel Realsense D455 cameras are mounted across from the robot to capture position of the object by color-thresholding pointcloud readings or retrieving pose estimation from aruco tags. Commands are sent to the controller at 5Hz. We restrict the end-effector workspace of the robot in a rectangle for safety so the robot arm doesn't collide dangerously with the table and objects outside

the workspace. We conduct extensive domain randomization and randomize the initial gripper pose during simulation training. The reward is computed from measured proprioception of the robot and estimated pose of the object. Details for each task are listed below.

**Hammering.** For hammering, the action is 3-dimensional and sets delta joint targets for 3 joints of the robot using joint position control. The observation space is 12-dimensional and includes end-effector cartesian xyz, joint angles of the 3 movable joints, joint velocities of the 3 movable joints, the z position of the nail, and the xz position of the goal. Each trajectory is 50 timesteps. In simulation, we randomize over the position, damping, height, radius, mass, and thickness of the nail. Details are listed in Tab. 1.

The reward function is parameterized as $r(t) = -10 \cdot r_{\text{nail}-\text{goal}}(t)$ where $r_{\text{nail}-\text{goal}} = (\mathbf{r}_{\text{nail}})_z - (\mathbf{r}_{\text{goal}})_z$ represents the distance in the z dimension of the nail head to the goal, which we set to be the height of the board the nail is on.

**Pushing.** For pushing, the action is 2-dimensional and sets delta cartesian xy position targets using end-effector position control. The observation space is 4-dimensional and includes end-effector cartesian xy and the xy position of the puck object. Each trajectory is 40 timesteps. In simulation, we randomize over the position of the puck. Details are listed listed in Tab. 3.

Let $\mathbf{r}_{\text{ee}}$ be the cartesian position of the end effector and $\mathbf{r}_{\text{obj}}$ be the cartesian position of the object. The reward function is parameterized as $r(t) = -r_{\text{ee}-\text{goal}}(t) - r_{\text{obj}-\text{goal}}(t) + r_{\text{threshold}}(t) - r_{\text{table}}(t)$ where $r_{\text{ee}-\text{goal}}(t) = \|\mathbf{r}_{\text{ee}}(t) - \mathbf{r}_{\text{obj}}(t) + [3.5\text{cm}, 0.0\text{cm}, 0.0\text{cm}]\|$ represents the distance of the end effector to the back of the puck, $r_{\text{obj}-\text{goal}}(t) = \|(\mathbf{r}_{\text{obj}}(t))_x - 55\text{cm}\|$ represents the distance of the puck to the goal (which is the edge of the table along the x dimension), $r_{\text{threshold}}(t) = \mathbb{I}[r_{\text{obj}-\text{goal}}(t) \geq 2.5\text{cm}]$ represents a goal reaching binary signal, and $r_{\text{table}}(t) = \mathbb{I}[(r_{\text{obj}}(t))_z \leq 0.0]$ represents a binary signal for when the object falls of the table.

**Inserting.** For inserting, the action is 3-dimensional and sets delta cartesian xyz position targets using end effector position control. The observation space is 9-dimensional and includes end-effector cartesian xyz, the xyz of the leg, and the xyz of the table hole. Each trajectory is 40 timesteps. We enlarge the table by a scale of 1.08 compared in the original table in FurnitureBench in order to make the table leg insertable without twisting. In simulation, we randomize the initial gripper position, position of the table, and friction of both the table and the leg.

Let $\mathbf{r}_{\text{pos1}}(t)$ and $\mathbf{r}_{\text{pos2}}(t)$ represent the Cartesian positions of the leg and table hole. Let:

$$x_{\text{distance}}(t) = \text{clip}\left(|\mathbf{r}_{\text{pos1},x}(t) - \mathbf{r}_{\text{pos2},x}(t)|, 0.0, 0.1\right)$$
$$y_{\text{distance}}(t) = \text{clip}\left(|\mathbf{r}_{\text{pos1},z}(t) - \mathbf{r}_{\text{pos2},z}(t)|, 0.0, 0.1\right)$$
$$z_{\text{distance}}(t) = \text{clip}\left(|\mathbf{r}_{\text{pos1},y}(t) - \mathbf{r}_{\text{pos2},y}(t)|, 0.0, 0.1\right)$$

Let the success condition be defined as:

$$r_{\text{success}}(t) = \mathbb{I}\left[x_{\text{distance}}(t) < 0.01 \text{ and } y_{\text{distance}}(t) < 0.01 \text{ and } z_{\text{distance}}(t) < 0.01\right]$$

The reward function is now:

$$r(t) = r_{\text{success}}(t) - 100 * \left(x_{\text{distance}}(t)^2 + y_{\text{distance}}(t)^2 + z_{\text{distance}}(t)^2\right)$$

**Sim2Sim Environment.** We additionally attempt to model a sim2real dynamics gap in simulation by taking the hammering environment and create a proxy for the real environment by fixing the domain randomization parameters, fixing the initial gripper pose, and rescaling the action magnitudes before rolling out in the environment.

## C  IMPLEMENTATION DETAILS

**Algorithm Details.** We use SAC as our base off-policy RL algorithm for training in simulation and fine-tuning in the real world. For our method, we additionally add in two networks: a dynamics model that predicts next state given current state and action, and a state-conditioned value network which regresses towards the Q-value estimates for actions taken by the current policy. These networks are training jointly with the actor and critic during SAC training in simulation.

**Network Architectures.** The Q-network, value network, and dynamics model are all parameterized by a two-layer MLP of size 512. The dynamics model is implemented as a delta dynamics model where model predictions are added to the input state to generate next states. The policy network produces the mean $\mu_a$ and a state-dependent log standard deviation $\log \sigma_a$ which is jointly

Table 1: Domain randomization of hammering task in simulation

| Name | Range |
|------|-------|
| Nail x position (m) | [0.3, 0.4] |
| Nail z position (m) | [0.55, 0.65] |
| Nail damping | [250.0, 2500.0] |
| Nail half height (m) | [0.02, 0.06] |
| Nail radius (m) | [0.005, 0.015] |
| Nail head radius (m) | [0.03, 0.04] |
| Nail head thickness (m) | [0.001, 0.01] |
| Hammer mass (kg) | [0.015, 0.15] |

Table 2: Domain randomization of pushing task in simulation

| Name | Range |
|------|-------|
| Object x position (m) | [0.0, 0.3] |
| Object y position (m) | [-0.25, 0.25] |

Table 3: Domain randomization of inserting task in simulation

| Name | Range |
|------|-------|
| Parts x/y position (m) | [-0.05, 0.05] |
| Parts rotation (degrees) | [0, 15] |
| Parts friction | [-0.01, 0.01] |

learned from the action distribution. The policy network is parameterized by a two-layer MLP of size 512, with a mean head and log standard deviation head on top parameterized by a FC layer.

**Pretraining in Simulation.** For hammering and puck pushing, we collect 25,000,000 transitions of random actions and pre-compute the mean and standard deviation of each observation across this dataset. We train SAC in simulation on the desired task by sampling 50-50 from the random action dataset and the replay buffer. We normalize our observations by the pre-computed mean and standard deviation before passing them into the networks. We additionally add Gaussian noise centered at 0 with standard deviation 0.004 to our observations with 30% probability during training. For inserting, we train SAC in simulation with no normalization. We train SAC with autotuned temperature set initially to 1 and a UTD of 1. We use Adam optimizer with a learning rate of $3 \times 10^{-4}$, batch size of 256, and discount factor $\gamma = .99$.

**Fine-tuning in Real World.** We pre-collect 20 real-world trajectories with the policy learned in simulation to fill the empty replay buffer. We then reset the critic with random weights and continue training SAC with a fixed temperature of $\alpha = 0.01$ and with a UTD of $2d$ with the pretrained actor and dynamics model. We freeze the value network learned from simulation and use it to relabel PBRS rewards during fine-tuning. During fine-tuning, for each state sampled from the replay buffer, we additionally hallucinate 5 branches off and add it to the training batch. As a result, our batch size effectively becomes 1536. The policy, Q-network, and dynamics model are all trained jointly on the real data during SAC fine-tuning. We don't train on any simulation data during real-world fine-tuning because we empirically found it didn't help fine-tuning performance in our settings.

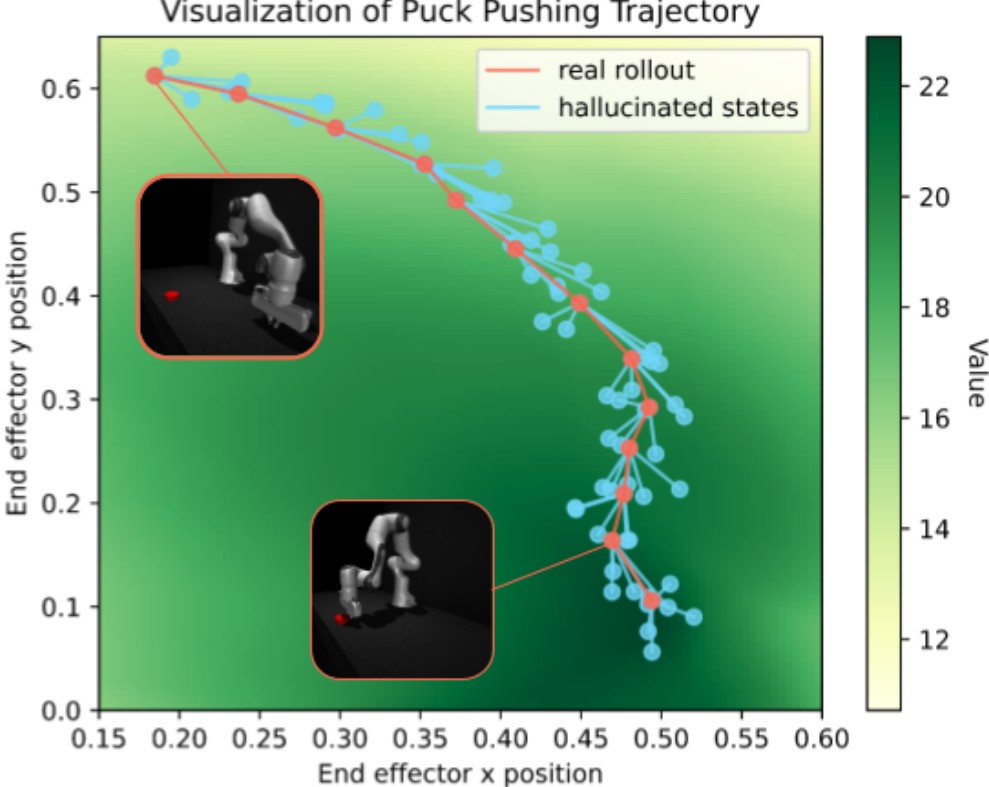

Figure 5: **Visualization of real rollout, hallucinated states, and value function.** The red dots indicate states along a real rollout in simulation. The blue dots indicate hallucinated states branching off real states generated by the learned dynamics model. The green heatmap indicates the value function estimates at different states. A corresponding image of the state is shown for two states. Since it is hard to directly visualize states and values due to the high-dimensionality of the state space, we only show a part of the trajectory where the puck does not move. This allows us to visualize states and values along changes in only end effector xy.

## D  QUALITATIVE RESULTS

We analyze the characteristics of hallucinated states and value functions in Fig. 5. We visualize a trajectory of executing puck pushing in simulation using the learned policy in this plot. The red dots indicate states along a real rollout in simulation. The blue dots indicate hallucinated states branching off real states generated by the learned dynamics model. The green heatmap indicates the value function estimates at different states. A corresponding image of the state is shown for two states. The trajectory shown in the figure shows the learned policy moving closer to the puck before pushing it. The value function heatmap shows higher values when the end effector is closer to the puck and lower values when further. Hallucinated states branching off each state show generated states for fine-tuning the learned policy.

Note that it is hard to directly visualize states and values due to the high-dimensionality of the state space. To get around this for puck pushing, we only show a part of the trajectory where the puck does not move. This allows us to visualize states and values along changes in only end effector xy.

# E    SIM-TO-SIM EXPERIMENTS

Here we additionally test each of the proposed methods on the sim-to-sim set-up from Du et al. (2021), which is meant to mock sim-to-real gaps but for familiar RL benchmark tasks. The results are depicted in Figure 7 for the Walker Walk, Cheetah Run, and Rope Peg-in-Hole environments. For all tasks, we use the precise settings from Du et al. (2021). Note that the general trend of these results matches our real world experiments – *SGFT* substantially accelerates learning and overcomes the dynamics gap between the 'simulation' and 'real' environments.

Finally, we additionally use the Walker and Peg-in-Hole environments to ablate the effects of the hyper parameter $H$. Intuitively, the peg-in-hole environment requires much more precise actions, and is thus should be more sensitive to errors in the pretraining environment. Thus, we should expect that SGFT will benefit from larger values of $H$ for this environment, as this will correspond to relying more heavily on returns from the target environment. We see that this trend holds in Figure 6, where the Walker environments is barely affected by the choice of $H$ but this hyper parameter has a large impact on the performance in the Peg-in-Hole Environment.

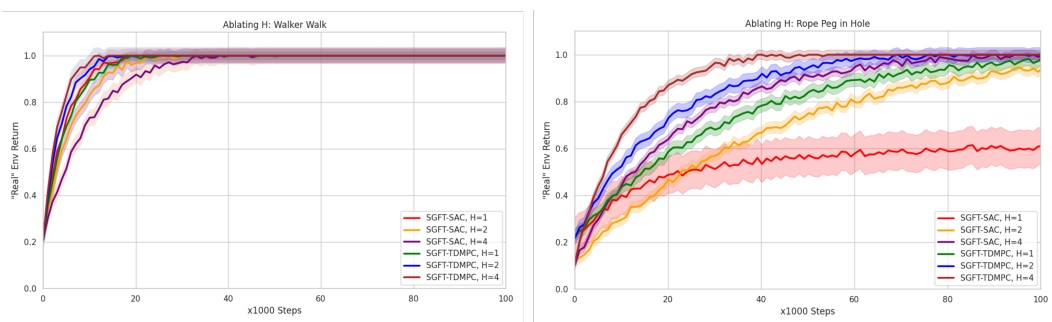

Figure 6: **Normalized Rewards for Sim-to-Sim Transfer**. Ablating the effects of the horizon $H$ across two sim-to-sim expirments. The choice of $H$ has a much larger effect on the peg-in-hole task, which requires much more precise actions to achieve succes.

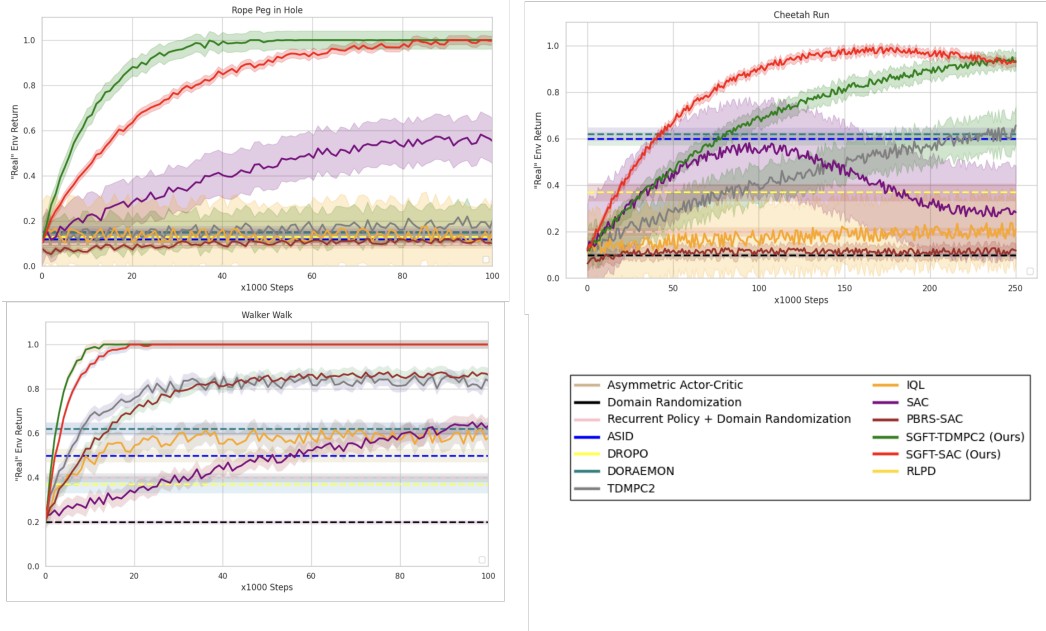

Figure 7: **Normalized Rewards for Sim-to-Sim Transfer**. We plot the normalized rewards for two sim-to-sim transfer tasks, where the rewards are normalized by the maximum reward achieved by any method.

