# OpenReview forum: "Rapidly Adapting Policies to the Real-World via Simulation-Guided Fine-Tuning"
_ICLR.cc/2025/Conference — ICLR 2025 Poster_

### Official Review · Reviewer_HzhY · 2024-11-03

**Soundness:** 3
**Presentation:** 2
**Contribution:** 2
**Rating:** 6
**Confidence:** 4

**Summary:**

This paper addresses the challenge of sim-to-real adaptation. It introduces a novel approach where the value function, learned in simulation, is used to reshape the reward signal, effectively guiding online reinforcement learning in real-world environments. The proposed method demonstrates superior performance compared to domain randomization and online system identification methods in three contact-rich and dynamic tasks: pushing, insertion, and hammering.

**Strengths:**

- The paper is focusing on an important problem.
 - The proposed method is novel, built on the key observation that the value function from a well-performing policy learned in simulation can be used to effectively reshape the reward for online learning.

**Weaknesses:**

- The introduction is hard to follow with the logic among paragraphs not clear.
 - The methodology and figure illustrations can be further improved.
   - Figure 2 is confusing. According to RL literature conventions, V_s in general, means the value function, but it is labeled as the world model in Figure 2.
   - How is the dynamic model learned? There is no description of learning the dynamics model in the main script or in the main algorithm, algorithm 1.

Typos:
 - Line 19 in abstract, “is to inefficient” -> ”is too inefficient”

**Questions:**

Please refer to the weakness section.

---

> ### Author Response · Authors · 2024-11-21
>
> Dear Reviewer,
>
> Thank you for your constructive comments! We have rewritten the introduction and added a small contributions section to clarify the framing of our approach. We have also updated our figures and writing to fix the errors that you have highlighted. Before addressing these points in more detail, we note that we have also included **two new deformable object manipulation tasks**. These are very difficult tasks for existing sim-to-real approaches, and the performance gains of SGFT are even more pronounced on these tasks. See the meta-comment above for details, and the ‘Rebuttal Figures’ in the supplementary for plots. We intend to add these new results to the paper draft soon.
>
> 1\. **Improving introduction**
>
> In the new draft of the introduction, the main point of each paragraph is as follows:
>
> i) Robot learning is a powerful paradigm, but collecting enough real-world data for current algorithms is challenging and expensive.
>
> ii) Physics simulators can provide a scalable, cheap source of off-domain (i.e. inaccurate) data.
>
> iii) However, due to the domain gap, current transfer techniques struggle for challenging contact-rich tasks.
>
> iv) Prior methods have investigated fine-tuning policies pretrained in simulation using real-world data, but use unstructured exploration strategies which are inefficient and impractical for real-world learning.
>
> v) In contrast, we argue for distilling information from the simulator that will guide and accelerate real-world fine-tuning.
>
> vi) In more detail, we accomplish this by reshaping objectives with value functions. We show this is particularly effective when paired with MBRL algorithms.
>
> Contributions:
>
> 1\) We introduce a light-weight framework which substantially accelerates existing fine-tuning approaches.
>
> 2\) We systematically show SGFT significantly outperforms numerous baselines across many real-world tasks.
>
> 3\) We provide theoretical analysis for the performance gains we observe.
>
> Please let us know if there is anything we can do to make our contribution clearer!
>
> 2\. **Improving figure illustrations**
>
> Thank you for pointing out the typo in Figure 2. We have fixed the notation, also modified the figure to make our framework easier to parse. Please let us know if you have any further suggestions!
>
> 3\. **Clarification on how dynamics model is learned**\
> The dynamics model is learned by fitting a generative model on the dataset, following the approach of conventional MBRL approaches such as MBPO and TD-MPC2. In our implementation, the dynamics model is trained via maximum likelihood to predict the next state given the current state and action. Please refer to Algorithm 2 (Dyna-SGFT) in the paper for further details on how we learn the dynamics model.
>
> 4\. **Typo in abstract**
>
> Thank you for pointing out the typo in the abstract. We have fixed “is to inefficient” -> ”is too inefficient”.\
> \
> We hope these additional explanations and modifications have addressed your previous questions. Please don’t hesitate to let us know for any additional comments or questions.

---

> > ### Author Response · Authors · 2024-12-02
> >
> > As the discussion phase draws to a close, we wanted to check in and see if our response has addressed your concerns and if there are any remaining points we can address.

---

### Official Review · Reviewer_TjP5 · 2024-11-03

**Soundness:** 2
**Presentation:** 3
**Contribution:** 3
**Rating:** 6
**Confidence:** 4

**Summary:**

The authors propose a novel framework (SGFT) for sim-to-real transfer in robot learning by finetuning RL policies pre-trained in simulation efficiently and effectively on limited real-world interactions. In particular, SGFT changes the real-world MDP problem by (1) turning it into a finite and short-horizon MDP (H-step) and (2) reshaping the new MDP's rewards with a pre-trained critic in simulation. These two changes allow the method to quickly adapt pre-trained policies with real-world interactions, and be also used in combination with model-based RL to further improve data efficiency.

**Strengths:**

- The paper uses a clear writing style, where the critical statements are repeated multiple times and highlighted.
- The method demonstrates good results on contact-rich real-world tasks---which are highly relevant to the community---in 100-200 real rollouts.
- Theoretical analysis of the proposed method is included
- The method is relatively simple to implement.

**Weaknesses:**

- Missing references to recent, related works: the authors discuss sim2real transfer works in the first paragraph of Sec. 2, but fail to include recent state-of-the-art methods as of 2023 and 2024 that "adapt simulation parameters to real-world data" (DROPO [1]), or that "learn adaptive policies to account for changing real-world dynamics" (DORAEMON [2]). I suggest skimming through these papers to make sure these two literature branches are well referenced and discussed.

-  Limited experimental evaluation: I raise concerns over the main empirical findings in Fig. 4.
	- The SAC baseline seems to perform overly bad. To my understanding, this baseline essentially starts the same way as SGFT, but then diverges as finetuning goes on in that SAC also changes its own critic, whereas SGFT relies on a frozen initial critic (cf. my question below for further details). Also, in light of recent claims [3], one would expect off-policy algos to perform better if correctly finetuned.
	- In my opinion, the comparison to DR baselines should be done by pre-training them with omniscent critic (aka asymmetric actor-critic), as described in [4] and often done in recent works [5]. SGFT must be trained in simulation with unpriviliged critics, hence it's making a subtle further assumption that DR methods can, instead, relax and leverage.
	- the Recurrent policy + DR baseline seems to be missing from Fig. 4 pushing. Pushing is also a notorious benchmark task that has been used by DORAEMON [2] and Peng et al. [4] to showcase successful zero-shot transfer in their experiments, by randomizing similar properties (e.g. mass, friction coefficient). How are zero-shot DR baselines performing in this setting?
	- Recent state-of-the-art zero-shot transfer methods such as Doraemon [2] should be included in the experimental evaluation.
    - Only experiments in the edge case H=1 are shown. This makes it hard to motivate the need for a more general framework.



[1] Tiboni, G., Arndt K., and Kyrki V. "DROPO: Sim-to-real transfer with offline domain randomization." Robotics and Autonomous Systems 166 (2023): 104432.

[2] Tiboni, G. et al. "Domain Randomization via Entropy Maximization." ICLR 2024.

[3] Ball, Philip J., et al. "Efficient online reinforcement learning with offline data." International Conference on Machine Learning. PMLR, 2023.

[4] Peng, Xue Bin, et al. "Sim-to-real transfer of robotic control with dynamics randomization." 2018 IEEE international conference on robotics and automation (ICRA). IEEE, 2018.

[5] Handa, Ankur, et al. "Dextreme: Transfer of agile in-hand manipulation from simulation to reality." 2023 IEEE International Conference on Robotics and Automation (ICRA). IEEE, 2023.

**Questions:**

- When H=1, what is the difference between a standard actor-critic (AC) method with a frozen critic vs. your SGFT method? To me, it seems that an AC method with frozen pre-trained critic essentially describes the SGFT method. If so, this could yield further intuitions and explanations of the algorithm. In other words, when H=1, it seems to me like the finetuning problem is turned into a contextual bandit problem where no sequential decision making is involved.

- I'm not convinced by the statement "optimizing Equation (3) guides policy optimization algorithms towards policies which increase the value of V_sim over the H-step windows." in Sec. 4.3. To my understanding, as H increases, the algorithm actually gives more importance to real-world returns rather than increasing the value of V_sim. In other words, a final state S_H with lower V_sim(S_H) could be preferred if the real H-step return to get to that state is relatively higher than it was in sim. Conversely, the statement gets truer the lower the value of H.

---

> ### Author Response · Authors · 2024-11-21
>
> Dear Reviewer,
>
> Thank you for your feedback on our work! We have been busy running the additional experiments requested by you and the other reviewers. We have uploaded all of the figures we will reference to a document titled ‘Rebuttal Figures’ attached in the supplementary; we will incorporate these into the paper draft in the coming days.  Please see our common response for a summary of the new experiments, including two new real-world deformable object manipulation tasks with a squishy ball and a towel. These are difficult tasks for sim-to-real methods wherein the benefits of SGFT are substantial.
>
> We have run the additional baselines you requested on the puck, ball, and cloth pushing tasks, **and are planning to also run them on the hammering and insertion tasks tomorrow**. However, we wanted to respond as soon as possible to your concerns with our current experiments, and give you a chance to let us know if there are any other points we can address in the coming days.
>
> 1\. **Adding references to state-of-the-art methods**
>
> We thank you for pointers to recent state-of-the-art methods: DROPO \[1] and DORAEMON \[2]. We have now discussed them in the updated version of the related work, with the following text:
>
> “Another appealing approach is to search for distributions of domain randomization parameters that will lead to maximally robust transfer in a principled fashion. For example, Tiboni et al. (2023a) uses entropy maximization to find simulator parameters which will lead to maximally robust transfer while accurately describing a small number of real world trajectories. This is taken further by Tiboni et al. (2023b), which automatically evolves training distributions purely in simulation to generate a curriculum of environments which leads to policies which are both performant and robust.”
>
> Moreover, we have included these baselines in our experiments for the puck, ball, and cloth pushing tasks. In these three tasks, DROPO and DORAEMON generally perform favorably compared to other direct transfer methods. However, with additional fine-tuning data SGFT is able to improve over the success rates of these methods. Please refer to Figures 1, 2, and 3 for further details. In future work, we would be very interested to see if these methods could provide a better initial policy for fine-tuning.
>
> 2\. **Clarification on performance of SAC baseline**
>
> We use the StableBaselines3 implementation of SAC, which is robust and widely used. Moreover, because SGFT-SAC is built on top of SAC, our method depends on the strength of the implementation, and we carefully tuned SAC for our experiments. This is further highlighted by the performance of SAC in our simulation experiments (Figure 10), where SAC generally performs comparably to SOTA methods such as TDMPC. Thus, we contend that the performance of our SAC implementation is reasonable and that learning in the real world is significantly harder for many methods than learning in simulation. Nonetheless, we ran RLPD \[3] on the puck, ball and cloth pushing tasks, and found that it led to only minimal improvements over our SAC implementation.
>
> 3\. **Additional experiments with asymmetric actor critic**
>
> We note that the policy from SGFT can actually be trained with asymmetric actor-critic in simulation to obtain high-performance policies. Then, one could estimate a critic for this policy using only available observations (i.e. no privileged information) to use with SGFT. Nonetheless, we agree that comparing to asymmetric actor-critic domain methods is important, and we have added this as a baseline to our hardware experiments. We found that it did not significantly impact performance for domain randomization methods on the puck, ball, and cloth pushing tasks. Please refer to Figures 1, 2, and 3 for details.
>
> 4\. **Evaluating all baselines for real-world pushing task**
>
> We evaluate all baseline methods we missed in the initial submission for the pushing tasks in the real-world: Recurrent Policy + Domain Randomization, ASID, TDMPC2, SGFT-TDMPC2, IQL, DROPO\[1], DOREAMON\[2], Asymmetric Actor-Critic\[3], and RLPD\[4]. We observe that both versions of SGFT substantially outperform prior fine-tuning methods, and yield better final policies than direct transfer methods. Please refer to Figure 1 for details.

---

> ### Author Response · Authors · 2024-11-21
>
> 5\. **Ablations on hyperparameter H:**
>
> We would like to clarify that we used H=1 for SGFT-SAC and H=4 for SGFT-TDMPC in the experiments in the initial submission. For the rebuttal, we have additionally performed a more extensive ablation on the effects of H on two standard sim-to-sim benchmark experiments: Walker and Rope Peg-in-Hole \[5] (Figure 13). Here, the sim2real dynamics gap is proxied as perturbations in simulator dynamics. Since the Rope Peg-in-Hole Environment requires significantly more precise motions than Walker to solve, this enables us to tackle the question: do tasks that require higher precision control require higher H, which in turn makes the policy transfer more difficult? We find that the performance of SGFT is relatively insensitive to the choice of H on Walker. However, SGFT requires larger H to achieve high-performance on the more precise Rope Peg-in-Hole task. We believe these experiments motivate the need for the more general framework. Please refer to Figures 8 and 9 for details.
>
> 6\. **Question 1: “When H=1…”**
>
> This is a very interesting point, and an interesting intuition! We have addressed this in the next comment.
>
> 7\. **Question 2: “I’m not convinced by the statement…”**
>
> Thank you for drawing this to our attention; we agree we can improve the wording here. What we meant to say here was:
>
> “For **small** values of H, optimizing Equation (3) guides policy optimization algorithms towards policies that increase the value of V\_sim over H-step windows. In the extreme case where  H=1, SGFT greedily attempts to increase the value of V\_sim each step. Conceptually, in this special case we have reduced the policy search problem to a contextual bandits problem, which is much easier to solve than the original infinite horizon objective. Intuitively, this is because the critic is frozen and no bootstrapping in the real-world is required. More generally, as H becomes larger the dependence on V\_sim is lessened while the dependence on the real returns increase.”
>
> We have updated the draft accordingly, please let us know if we have cleared this point up!
>
> Thank you again for your feedback, and please let us know if there are further ways you believe we can improve our submission!
>
> \[1] Tiboni, G., Arndt K., and Kyrki V. "DROPO: Sim-to-real transfer with offline domain randomization." Robotics and Autonomous Systems 166 (2023): 104432.
>
> \[2] Tiboni, G. et al. "Domain Randomization via Entropy Maximization." ICLR 2024.
>
> \[3] Ball, Philip J., et al. "Efficient online reinforcement learning with offline data." International Conference on Machine Learning. PMLR, 2023.
>
> \[4] J. Zhang, M. Heo, Z. Liu, E. Biyik, J. J. Lim, Y. Liu, and R. Fakoor, “Extract: Efficient policy learning by extracting transferrable robot skills from offline data,” arXiv preprint arXiv:2406.17768, 2024.

---

> > ### Comment · Reviewer_TjP5 · 2024-11-24
> >
> > Thank you for the prompt implementation of the new experiments. I appreciate the new tasks the authors have implemented during the rebuttal phase, as well as the ablation on the H hyperparameter.
> >
> > Minor concerns:
> > 1.  As far as I understand, DROPO does not use entropy maximization, this is only introduced in DORAEMON. You may want to adjust the new phrase you added when introducing these two works.
> > 2.  Fig. 4 still seems to be missing numerous baselines for the hammering and insertion tasks. Will these be added in the future?
> > 3. In Sec. 6.1, " we set the ‘done’ flag to true at the end of each rollout.". For H=1, does this mean after every state transition? Since this comes right after saying that you set H=1 across all experiments, I think it should be clarified more explicitly. Also, I tend to forget about this implementative aspect when thinking of your method, but it's a rather substantial detail. I'd suggest to highlight it, as this is really what turns the problem into a contextual bandit setting.

---

> > > ### Author Response · Authors · 2024-11-25
> > >
> > > Thank you for reading our responses carefully\!
> > >
> > > **Minor Concerns**
> > >
> > > 1. Thanks for pointing out this inaccuracy. The paper now reads:
> > >
> > > “Another appealing approach is to search for distributions of domain randomization parameters that will lead to maximally robust transfer in a principled fashion. For example, Tiboni et al. (2023a) uses a small pre-collected data set to estimate distributions over dynamics parameters and improve robustness, rather than collapsing to a point estimate as in pure system identification. This is taken further by Tiboni et al. (2023b), which uses an entropy maximization to strategy to automatically evolve training distributions purely in simulation to generate a curricula of environments which leads to polices which are both performant and robust.”
> > >
> > > We meant to say that DROPO’s robustness properties come from the fact that it learns a distribution of dynamics parameters with non-trivial entropy (as in point estimates from system ID), but agree calling it an \`entropy maximization’ strategy was inaccurate.
> > >
> > > ----
> > > 2. We have now finished all of the benchmarks on the hammering and insertion tasks. The plots are both in the updated version of the paper (Fig 4\) and the rebuttal document (Figs 14 \+ 15). The rough trends reported for the new baselines on pushing hold for these environments.
> > >
> > > ----
> > > 3. You are correct. We have attempted to make this more explicit:
> > >
> > > “We use H=1 in all our hardware experiments. To implement our approach, we set the \`done' flag to true at the end of each roll-out. In the case where H=1, this means applying \`dones’ at the end of each transition. As discussed above, this is equivalent to reducing the problem to a contextual bandits problem, which is a much easier problem to solve than the original infinite horizon objective. In effect, we freeze the critic learned in simulation, only updating the policy to rapidly adapt to dynamics shifts.”
> > >
> > > ----
> > >
> > > Please let us know if this addresses your remaining concerns, and if there is anything else we can address in the last few days\!

---

> ### Comment · Reviewer_TjP5 · 2024-11-26
>
> I thank the authors for the clear responses to all my concerns. The current last revision of the paper displays a thorough experimental evaluation, which makes the paper a valuable addition to the field. I'm therefore keeping the score as six as I think it's worth accepting it. I'm not raising the score higher than six as I believe the method could benefit from a better theoretical explanation of the implication to the RL problem when reducing the finetuning problem to a contextual bandit problem. I'm suspecting there could be connections to existing works and settings that are missing, as the practical implementation of the method is rather straightforward and simple (i.e. frozen critic and setting `done=True` for each transition). For example, could this approach be connected to "residual learning" [1,2]? E.g. in [2] they mention:
> > Every time a new task needs to be learned, we can transfer our learned residuals and learn the component of the Q-function that is task-specific, hence, maintaining the task structure from prior behaviors.
>
> Unfortunately I don't have enough theoretical expertise to judge for myself and hence will leave my score as it is.
>
> [1] Klink, Pascal, et al. "Boosted curriculum reinforcement learning." International Conference on Learning Representations. 2021.
>
> [2] Jauhri, Snehal, Jan Peters, and Georgia Chalvatzaki. "Robot learning of mobile manipulation with reachability behavior priors." IEEE Robotics and Automation Letters 7.3 (2022): 8399-8406.

---

### Official Review · Reviewer_q12p · 2024-11-04

**Soundness:** 3
**Presentation:** 3
**Contribution:** 2
**Rating:** 6
**Confidence:** 4

**Summary:**

The paper proposes a method that learns a robust transferred policy in sim-to-real settings by learning a value function, and then fixing it for H-step MPC. The deployed policy only needs to learn to optimize H step model rollouts with the assumption that even if the low level dynamics differ between simulation and the real world, the value function through temporal extension of H-steps is more reliable in deciding which are the target future states in the trajectory. The authors demonstrate both RL fast policy and MPC optimization based variants of the actor trained to optimize H step trajectories in the real environment. The method is tested on three simulation-to-real robot manipulation tasks.

**Strengths:**

1. The method is simple and elegant, and the motivation for doing H step planning is clear.
2. The assumption that the low level actions learnt in simulation can be incorrect, but the general value difference between states preserved is a reasonable assumption, and verified in the experiments.
3. The value functions being frozen and just finetuning the policy is likely significantly more sample efficient than finetuning all components.
3. The demonstration of TDMPC2 type model learning and planning in real world robot experiments is a great addition.

**Weaknesses:**

1. The choice of hyperparameter H seems extremely important, and no ablations have been done here. All algorithms have hyperparameters, but atleast the most important ones should have ablation experiments.
2. Following up on the previous point, a major weakness of the paper is the lack of extensive experiments. Considering that the proposed method is mainly an engineering improvement rather than a wholly unique algorithm, more experiments in other real world and simulated domains could have been appreciated. For example, within simulated domains there could have been experiments where the model is trained on perturbed dynamics (such as different mass of limbs for locomotion agents) and then transferred to an unperturbed environment. Other experiments could have been designed to show how the optimal values of H are related to different task types. For example it would be interesting to see if tasks that require higher precision control require higher H, which in turn makes the policy transfer more difficult.
3. There is not much novelty in the method itself, and it might even be something that people already kind of do. Especially SGFT-SAC, the authors fix H=1 for the experiments and this just means that the value function is fixed after simulation and just the policy is finetuned. The TDMPC experiments are more interesting, but this is not then evaluated in the pushing task.

**Questions:**

Questions:
1. In Figure 4, why are many of the baseline methods not evaluated for the pushing task?
2. What happens if you use smaller H for SGFT-TDMPC? What about larger H?

---

> ### Author Response · Authors · 2024-11-21
>
> Dear Reviewer,
>
> Thank you for your feedback on how we can improve our evaluation to better justify our claims. We have performed 7 new real-world experiments and simulated ablations to address your concerns. Then we seek to clarify how SGFT, while simple to implement, is a substantial conceptual departure from prior fine-tuning methods such as \[6] and \[7].
>
> We have uploaded all of the figures we will reference to a document titled ‘Rebuttal Figures’ attached in the supplementary; we will incorporate these into the paper draft in the coming days.
>
> 1\. **More extensive real-world and simulation experiments**
>
> We have included 5 new hardware experiments, including deformable object manipulation tasks, and 2 new sim-to-sim experiments to further highlight the capabilities of SGFT.
>
> 1.1. **Real-world deformable object manipulation tasks**
>
> We add two additional real-world tasks, where the objective is to push deformable objects (a towel and a squishy toy ball) to a desired location (Figures 11 and 12). Deformable objects are notoriously difficult to model, and a significant challenge for sim-to-real methods. To achieve this, we train a pushing value function in simulation on a small range of objects with varying physical parameters (size, mass, etc). Even though the distribution of simulated objects is extremely different from the real-world deformable objects, SGFT achieves a near 100% success rate in 100 trajectories. Meanwhile, for these substantially harder tasks, all other baselines achieve less than 50% success rate for the toy ball and less than 30% for the towel. Details depicted in Figures 2 and 3.
>
> 1.2. **Additional sim-to-sim experiments**
>
> In addition to the two standard sim-to-sim benchmark experiments (Walker and Cheetah) from \[5] in Appendix E of the initial submission, we have added an additional environment: Rope Peg-in-Hole. In each of these experiments, SGFT substantially outperforms baseline transfer and fine-tuning methods. Details depicted in Figures 5, 6, and 7.
>
> 1.3. **Ablations on hyperparameter H**
>
> We ablate over the choice of H on two standard sim-to-sim benchmark experiments: Walker and Rope Peg-in-Hole \[5] (Figure 13). Here, the sim2real dynamics gap is proxied as perturbations in simulator dynamics. Since the Rope Peg-in-Hole Environment requires significantly more precise motions than Walker to solve, this enables us to tackle the question: do tasks that require higher precision control require higher H, which in turn makes the policy transfer more difficult?
>
> We find that the performance of SGFT is relatively insensitive to the choice of H on Walker. However, as the reviewer conjectured, SGFT requires larger H to achieve high-performance on for Rope Peg-in-Hole. Please refer to Figures 8 and 9 for details. We thank the reviewer for suggesting this sort of experiment!
>
> 1.4. **(New) baseline methods for pushing task**
>
> To address _question (1)_, we evaluate all baseline methods we missed in the initial submission for the pushing task in the real-world: Recurrent Policy + Domain Randomization, ASID, TDMPC2, SGFT-TDMPC2, IQL. Per request of reviewer TjP503, we additionally run baselines of DROPO\[1], DOREAMON\[2], Asymmetric Actor-Critic\[3], and RLPD\[4] on the pushing tasks. As with our previous experiments, SGFT substantially outperforms prior methods for these tasks. Please refer to Figures 1, 2, and 3 for details. We will also add the new baselines for hammering and insertion tomorrow.
>
> 1.5. **Additional real-world ablation on domain randomization**
>
> Per request from reviewer ZMTP07, we have additionally ablated the effects of domain randomization on our real-world experiments. In particular, we used SGFT to finetune policies which were trained without dynamics randomization. While this substantially reduced the effectiveness of the initial policies, it had a minimal effect on the fine-tuning performance of SGFT, demonstrating our method is insensitive to this hyper parameter. Details depicted in Figure 4.

---

> ### Author Response · Authors · 2024-11-21
>
> 2\. **SGFT is a paradigm shift for sim-to-real fine-tuning:**
>
> We argue that SGFT, while simple to implement, is a substantial paradigm shift from prior sim-to-real fine-tuning approaches such as \[6]\[7] and not just a minor engineering improvement. In short, these approaches a) pretrain a policy in simulation and b) simply use this as an _initialization_ for real world learning. This approach _does not retain the structure of the simulator during fine-tuning_. By transferring knowledge with the value function through reward shaping, SGFT is a novel, conceptually distinct paradigm which _leverages the structure of the simulator to guide and accelerate real-world learning_. We emphasize is a broad algorithmic insight and not simply an engineering trick for a particular setting - it is applicable across fine-tuning algorithms and tasks, and is distinct from what people currently do for sim to real fine-tuning. Moreover, we demonstrate it has theoretical grounding and leads to substantial empirical gains.
>
> We hope these additional explanations and modifications have addressed your previous questions. Please don’t hesitate to let us know for any additional comments or questions.
>
> \[1] Tiboni, G., Arndt K., and Kyrki V. "DROPO: Sim-to-real transfer with offline domain randomization." Robotics and Autonomous Systems 166 (2023): 104432.
>
> \[2] Tiboni, G. et al. "Domain Randomization via Entropy Maximization." ICLR 2024.
>
> \[3] Ball, Philip J., et al. "Efficient online reinforcement learning with offline data." International Conference on Machine Learning. PMLR, 2023.
>
> \[4] Peng, Xue Bin, et al. "Sim-to-real transfer of robotic control with dynamics randomization." 2018 IEEE international conference on robotics and automation (ICRA). IEEE, 2018.
>
> \[5] Yuqing Du, Olivia Watkins, Trevor Darrell, Pieter Abbeel, and Deepak Pathak. Auto-tuned sim-toreal transfer. In 2021 IEEE International Conference on Robotics and Automation (ICRA), pp. 1290–1296. IEEE, 2021.
>
> \[6] Marvin Zhang, Sharad Vikram, Laura Smith, Pieter Abbeel, Matthew Johnson, and Sergey Levine. Solar: Deep structured representations for model-based reinforcement learning. In International
>
> conference on machine learning, pp. 7444–7453. PMLR, 2019.
>
> \[7] Yunchu Zhang, Liyiming Ke, Abhay Deshpande, Abhishek Gupta, and Siddhartha S. Srinivasa.
>
> Cherry-picking with reinforcement learning. In RSS, 2023

---

> > ### Comment · Reviewer_q12p · 2024-11-25
> >
> > While I am still not fully convinced about this method being a distinct paradigm for sim-to-real as the authors state, I appreciate the detailed response and additional experiments/ablations. I am raising my score by 1 point and think the paper is worth accepting, primarily because of the extensive experiments.

---

> > > ### Author Response · Authors · 2024-11-25
> > >
> > > Thank you for considering our response! Please let us know if anything else comes to mind in the last few days.

---

### Official Review · Reviewer_ZMTP · 2024-11-07

**Soundness:** 3
**Presentation:** 2
**Contribution:** 2
**Rating:** 5
**Confidence:** 2

**Summary:**

This paper presents “Simulation-Guided Fine-Tuning” (SGFT), a framework aimed at improving sim-to-real adaptation by leveraging simulation-guided data augmentation and model-based reinforcement learning techniques. The paper uses real-world tasks like hammering, inserting, and puck pushing to test SGFT’s ability to adapt policies from simulation to reality with limited real-world data. The framework builds on a model-based RL backbone, specifically using SAC and TDMPC, to optimize policy adaptation with efficient data usage and a reward shaping mechanism.

**Strengths:**

- The paper addresses the sim-to-real gap in reinforcement learning, an essential area in robotics research. Using a model-based framework with potential-based reward shaping provides an innovative approach to sim-to-real transfer.
- SGFT’s modular design, integrating model-based learning with reward shaping, shows the potential for efficient adaptation without extensive real-world training. The authors provide theoretical analyses
- The evaluation on dynamic, real-world tasks such as hammering and inserting presents a realistic testing ground for examining SGFT’s efficacy.

**Weaknesses:**

1. Limited Task Scope and Complexity:
The evaluation primarily involves simple, structured tasks (hammering, inserting, and puck pushing), which limits the generalizability of the results. The tasks do not fully explore SGFT’s adaptability across a broader spectrum of real-world challenges or more complex environments. For example, the hammering task, despite involving contact dynamics, still relies on predefined, straightforward goals and does not include diverse object types or more intricate multi-step interactions. This narrow task set could mean that SGFT’s advantages may not scale effectively to more complex tasks .
2. Baseline Comparison and Performance:
While SGFT demonstrates some improvement in sample efficiency and asymptotic performance over baselines like SAC and domain randomization, the gains over TDMPC-2 and PBRS are limited. In Table 1, while SGFT performs comparably to these baselines in most cases, it does not consistently outperform them by a significant margin, raising questions about whether the added complexity of the SGFT approach is justified by these incremental improvements  .
3. Heavy Dependence on Simulation Accuracy and Engineering Effort:
SGFT’s approach requires a highly accurate simulation setup to guide policy learning effectively, which can be labor-intensive. The need for extensive domain randomization (e.g., Table 1 and 2 for hammering and puck pushing parameters) and environment-specific configurations could make SGFT less practical for tasks where precise dynamics modeling is challenging, such as interactions with liquids or deformable objects. This limitation suggests SGFT may struggle in scenarios where real-world dynamics differ significantly from the simulation environment  .
4. Uncertain Performance on More Complex Dynamics:
While the paper effectively adapts to straightforward tasks, its approach may not handle more nuanced dynamics. For instance, tasks involving materials with variable properties (like deformable objects or variable friction surfaces) may introduce unpredictable behaviors that SGFT, as presented, is unlikely to manage effectively. Thus, the scope of SGFT’s generalizability and flexibility across a broader range of robotics tasks remains unclear  .

**Questions:**

1. How would SGFT handle tasks involving more complex dynamics, such as fluid or deformable object manipulation, where precise dynamics modeling is difficult? Would this require substantial engineering modifications, or does SGFT have inherent limitations in these scenarios?
2. Can the authors provide insights into the expected engineering effort for adapting SGFT to new environments? Given the dependency on extensive domain randomization and parameter tuning, how feasible is it to scale SGFT across a diverse set of real-world tasks?
3. Are there specific plans to expand the evaluation to include more complex tasks or different robotic systems? Given the limited range of tasks tested, a broader evaluation could provide a more comprehensive understanding of SGFT’s robustness and adaptability.

---

> ### Author Response · Authors · 2024-11-21
>
> Dear Reviewer,
>
> Thank you for your thorough feedback on how we can improve the evaluation of our method. We have conducted hardware experiments for **two new deformable object manipulation tasks** to further highlight the capabilities of SGFT. Our central claim is that SGFT bypasses the need for an accurate simulator, reducing engineering effort and enabling transfer with substantially less fine-tuning data for tasks which are impractical to model accurately.
>
> For ease of access, we have uploaded all of the figures we will reference to a document titled ‘Rebuttal Figures’ attached in the supplementary; we will incorporate these into the paper draft in the coming days.
>
> 1\. **Scaling to more complex (deformable) tasks**
>
> To address _weaknesses (1)(4)_ and _question (1)(3)_, we add two additional real-world tasks, where the objective is to push deformable objects (a towel and a squishy toy ball) to a desired location (Figures 11 and 12). To achieve this, we train a pushing value function in simulation on a small range of objects with varying physical parameters (size, mass, etc). Even though the distribution of simulated objects is extremely different from the real-world deformable objects, SGFT achieves a near 100% success rate in 100 trajectories. Meanwhile, for these tasks, all other baselines achieve less than 50% success rate for the toy ball and less than 30% for the towel. Details depicted in Figures 2 and 3. This highlights our argument that value functions trained in simulation can provide impactful guidance and massively accelerate real-world learning even when the simulator is extremely inaccurate, reducing the need for engineering effort.
>
> All in all, we now have 5 real-world tasks, which we believe to be a reasonable spread over tasks of interest to robot learning. For example, the peg insertion task is a multi-step task, while hammering displays highly dynamic contact-rich motion. We believe this provides a broad evaluation which provides a more comprehensive understanding of SGFT’s robustness and adaptability.
>
> 2\. **SGFT substantially outperforms prior methods**
>
> To address _weakness (2)_, we would like to clarify that our primary claim is that SGFT boosts sample efficiency, not final performance. Across the board, SGFT is more sample efficient than prior methods, upwards of 2-3x on hammering, 2x on puck pushing, and even larger margins on the deformable tasks where prior methods struggle completely. Details are shown in Figures 1, 2, and 3. We view the fact that SGFT has higher final performance on some tasks to be an additional benefit, but is not the primary claim we aim to highlight.
>
> 3\. **SGFT requires minimal engineering overhead**
>
> To address _weakness (3)_ and _question (2)_, we first remark that our core reward-shaping techniques are very simple to implement, and a simple addition to existing sim-to-real pipelines. However, we also argue that it requires substantially less engineering effort when designing simulation environments compared to prior sim-to-real approaches. To further address this point, we additionally ran ablations for the puck pushing tasks where we **removed all dynamics randomization from the pretraining phase.** As depicted in Figure 4, this had barely any effect on the learning performance of SGFT. Thus, carefully tuning simulation parameters is not a particularly important ingredient of SGFT. Instead the methodology introduced in SGFT actually allows the fine-tuning procedure to overcome significant misspecification during simulator design, making the engineering task easier.
>
> We hope these additional explanations and modifications have addressed your previous questions. Please don’t hesitate to let us know for any additional comments or questions.

---

> > ### Author Response · Authors · 2024-12-02
> >
> > As the discussion phase draws to a close, we wanted to check in and see if our response has addressed your concerns and if there are any remaining points we can address.

---

### Author Response · Authors · 2024-11-21
**Meta Comment**

Here we would like to summarize how we have addressed the primary concerns raised by reviewers. We have conducted a wide range of new experiments, including **two new real-world experiments with deformable objects,** running additional baselines, and providing ablations on design decisions for our method. We also aim to address questions about the novelty, performance, and engineering effort required for our method.

1\. **New deformable object manipulation tasks**

Reviewers **ZMTP07** and **q12p03** asked for additional tasks. Reviewer **ZMTP07** specifically asked whether our approach would be applicable to harder tasks such as deformable object manipulation, which are extremely difficult to model. We have included **two new deformable object manipulation tasks** -- pushing both a towel and a squishy toy ball to desired locations (Figures 11 and 12). As noted by Reviewer **ZMTP07,** deformables are extremely difficult to model accurately, and very challenging for sim-to-real approaches. We demonstrate that SGFT scales to these more complex tasks, where it outperforms baselines by an even wider margin than on the tasks in the original submission. For example, on the cloth pushing task SGFT achieves a near 100% success rate on both new tasks in just 100 trials, while the next-best baseline only achieves 50% on the ball task and 30% for cloth.

2\. **Additional Baselines and simulation ablations**

Reviewer **TjP503** and **q12p03** asked for evaluations of additional baseline methods. We evaluate all baseline methods we missed in the initial submission for the pushing task in the real-world: Recurrent Policy + Domain Randomization, ASID, TDMPC2, SGFT-TDMPC2, IQL. We additionally run baselines of DROPO, DOREAMON, Asymmetric Actor-Critic, and RLPD on the pushing tasks. SGFT substantially outperforms prior methods for these tasks. We will also add the new baselines for hammering and insertion tomorrow.

Reviewer **q12p03** asked for additional sim-to-sim experiments. In addition to Walker and Cheetah environments from Appendix E of the initial submission, we have added the Rope Peg-in-Hole environment. In each of these experiments, SGFT substantially outperforms baseline transfer and fine-tuning methods.

Reviewers **q12p03** and **TjP503** asked for ablations over H. We ablate over the choice of H sim-to-sim on Walker and Rope Peg-in-Hole. We find that the performance of SGFT is relatively insensitive to the choice of H on Walker, while SGFT requires larger H to achieve high-performance on the more precise Rope Peg-in-Hole task. We believe these experiments motivate the need for the more general framework.

Reviewer **ZMTP07** raised the concern: “The need for extensive domain randomization (e.g., Table 1 and 2 for hammering and puck pushing parameters) and environment-specific configurations could make SGFT less practical for tasks where precise dynamics modeling is challenging.” To address this, we used SGFT to finetune policies in the real-world which were trained without dynamics randomization. While this substantially reduced the effectiveness of the initial policies, it had a minimal effect on the fine-tuning performance of SGFT, demonstrating our method is insensitive to this hyperparameter.

3\. **Novelty of SGFT**

Some reviewers raised questions about the novelty of our approach. To the best of our knowledge, SGFT proposes **a fundamentally new paradigm for sim-to-real fine-tuning** compared to prior fine-tuning approaches such as \[1]\[2]. In short, these approaches a) pretrain a policy in simulation and b) simply use this as an _initialization_ for real world learning. This approach _does not retain the structure of the simulator during fine-tuning_. By transferring knowledge with the value function through reward shaping, SGFT is a novel, conceptually distinct paradigm which _leverages the structure of the simulator to guide and accelerate real-world learning_. We emphasize is a broad algorithmic insight and not simply an engineering trick for a particular setting - it is applicable across different fine-tuning algorithms and tasks, and is distinct from what people currently do for sim to real fine-tuning. Moreover, we demonstrate it has theoretical grounding and leads to substantial empirical gains for two instantiations of the method, when compared to numerous baselines.

---

> ### Author Response · Authors · 2024-11-21
>
> 4\. **Engineering effort of SGFT**
>
> Reviewer **ZMTP07** raised the concerns over the engineering effort required for our method, specifically:  “SGFT’s approach requires a highly accurate simulation setup to guide policy learning effectively, which can be labor-intensive.”
>
> Our central claim is that SGFT bypasses the need for an accurate simulator, reducing engineering effort and enabling transfer with substantially less fine-tuning data for tasks which are impractical to model accurately. Moreover, our core reward-shaping insights are very simple to implement, and a straightforward addition to existing sim-to-real pipelines. To demonstrate that SGFT reduces the effort needed to design simulation environments, we additionally ran ablations for the puck pushing task where we **removed all dynamics randomization from the pretraining phase.** As depicted in Figure 4, this had barely any effect on the learning performance of SGFT. Thus, carefully tuning simulation parameters is not a particularly important ingredient of SGFT. Instead the methodology introduced in SGFT actually allows the fine-tuning procedure to overcome significant misspecification during simulator design, making the engineering task easier.
>
> 5\. **SGFT substantially outperforms prior methods**
>
> Reviewer **ZMTP07** raised concerns over the performance gains of SGFT over baselines: “SGFT performs comparably to these baselines in most cases, it does not consistently outperform them by a significant margin, raising questions about whether the added complexity of the SGFT approach is justified by these incremental improvements”
>
> We would like to clarify that our primary claim is that SGFT boosts sample efficiency, not final performance. Across the board, SGFT is more sample efficient than prior methods, upwards of 2-3x on hammering, 2x on puck pushing, and even larger margins on the deformable tasks where prior methods struggle completely. We view the fact that SGFT has higher final performance on some tasks to be an additional benefit, but is not the primary claim we aim to highlight.
>
> \[1] Marvin Zhang, Sharad Vikram, Laura Smith, Pieter Abbeel, Matthew Johnson, and Sergey Levine. Solar: Deep structured representations for model-based reinforcement learning. In International Conference on Machine Learning, pp. 7444–7453. PMLR, 2019.
>
> \[2] Yunchu Zhang, Liyiming Ke, Abhay Deshpande, Abhishek Gupta, and Siddhartha S. Srinivasa. Cherry-picking with reinforcement learning. In RSS, 2023

---

### Meta-Review · Area_Chair_vc7G · 2024-12-21

**Metareview:**

This paper introduces Simulation-Guided Fine-Tuning (SGFT), a framework for efficiently adapting simulation-trained policies to real robots by leveraging value functions learned in simulation as guidance for real-world exploration. The method combines short-horizon optimization with reward shaping to enable sample-efficient learning despite irresolvable sim-to-real gaps.

Main strengths:
- A fresh perspective to leverage simulation for real-world robot learning
- A new technical method that effectively leverages simulation value functions to guide real-world exploration. The idea of reward reshaping using V_sim is particularly elegant.
- Comprehensive empirical validation demonstrating 2-3x sample efficiency gains over baselines
- Clear presentation with theoretical backing for the approach

Limitations:
- Theoretical analysis could make stronger connections to existing work
- Some comparisons to recent baselines were added late in the review process

**Additional Comments On Reviewer Discussion:**

Key discussion points
- The authors demonstrated through ablation studies that SGFT's improvements come from both the architectural design and value function guidance, not just from having more training data
- Initial concerns about engineering overhead were addressed by showing SGFT actually reduces required setup effort
- Questions about real-world applicability were resolved through new experiments with deformable objects and practical inference speeds (~200 fps)
- Technical questions about horizon length selection and implementation details were addressed with additional ablations

The authors have been highly responsive throughout the discussion period.

---

### Decision · Program_Chairs · 2025-01-22

Accept (Poster)